# LMO2 is essential to maintain the ability of progenitors to differentiate into T-cell lineage in mice

Ken-ichi Hirano[1†], Hiroyuki Hosokawa[1,2†], Maria Koizumi[1], Yusuke Endo[3,4], Takashi Yahata[2,5], Kiyoshi Ando[2,6], Katsuto Hozumi[1]*

[1]Department of Immunology, Tokai University School of Medicine, Isehara, Japan; [2]Institute of Medical Sciences, Tokai University, Isehara, Japan; [3]Laboratory of Medical Omics Research, Kazusa DNA Research Institute, Kisarazu, Japan; [4]Department of Omics Medicine, Graduate School of Medicine, Chiba University, Chiba, Japan; [5]Department of Innovative Medical Science, Tokai University School of Medicine, Isehara, Japan; [6]Department of Hematology and Oncology, Tokai University School of Medicine, Isehara, Japan

*For correspondence:
hozumi@is.icc.u-tokai.ac.jp

†These authors contributed equally to this work

Competing interests: The authors declare that no competing interests exist.

**Abstract** Notch signaling primarily determines T-cell fate. However, the molecular mechanisms underlying the maintenance of T-lineage potential in pre-thymic progenitors remain unclear. Here, we established two murine *Ebf1*-deficient pro-B cell lines, with and without T-lineage potential. The latter expressed lower levels of *Lmo2*; their potential was restored via ectopic expression of *Lmo2*. Conversely, the CRISPR/Cas9-mediated deletion of *Lmo2* resulted in the loss of the T-lineage potential. Introduction of *Bcl2* rescued massive cell death of Notch-stimulated pro-B cells without efficient LMO2-driven *Bcl11a* expression but was not sufficient to retain their T-lineage potential. Pro-B cells without T-lineage potential failed to activate *Tcf7* due to DNA methylation; *Tcf7* transduction restored this capacity. Moreover, direct binding of LMO2 to the *Bcl11a* and *Tcf7* loci was observed. Altogether, our results highlight LMO2 as a crucial player in the survival and maintenance of T-lineage potential in T-cell progenitors via the regulation of the expression of *Bcl11a* and *Tcf7*.

## Introduction

Hematopoietic stem cells (HSCs) have the ability to self-renew and differentiate into all blood cell types. HSCs begin to differentiate into various lineage cells, gradually losing their potential to become their descendants, hematopoietic progenitor cells (HPCs), and then differentiate into mature blood cells (*Doulatov et al., 2012*; *Kosan and Godmann, 2016*). This sequence of processes has long been imagined as a ball rolling down a valley track (*Goldberg et al., 2007*). Cell fate decisions in hematopoietic cells are controlled by continuous interactions between environmental influences and intrinsic cellular mechanisms, such as transcription factor networks and epigenetic regulation (*Wilson et al., 2009*).

In the process of specification from HPCs, only the T-cell lineage requires a specialized environment of the thymus, where the immigrant cells receive Notch signaling induced by the interaction of Notch1 on the immigrant cells and a Notch ligand, delta-like 4 (Dll4), on the thymic epithelium, which determines their fate to the T-cell lineage (*Radtke et al., 1999*; *Hozumi et al., 2008*; *Koch et al., 2008*). In addition, several transcription factors contribute to the acquisition of T-cell identity, including TCF1 (encoded by *Tcf7*), basic helix-loop-helix (bHLH) factors, E2A and HEB, PU.1, GATA3, Bcl11b, and Runx family members, all of which are essential for T-cell development in the thymus (*Hosokawa and Rothenberg, 2021*; *Hosokawa et al., 2021a*; *Hosokawa et al., 2021b*).

Among them, one of the earliest Notch-activated genes, *Tcf7*, plays a critical role in Notch-mediated initiation of the T-lineage program (*Weber et al., 2011*; *Johnson et al., 2018*). The coordinated action of these environmental and intrinsic factors completes the commitment to T-cell lineage in double-negative (DN; CD4$^-$CD8$^-$) thymocytes at the transition from the DN2 to DN3 stages (*Yui and Rothenberg, 2014*; *Hosokawa and Rothenberg, 2021*). However, it remains unclear how the potential of the thymic immigrant cells is maintained to initiate their differentiation program toward the T-cell lineage.

*LMO2* gene is located near the breakpoint of the chromosomal translocation t(11;14) (p13;q11) in human T-ALL (*Boehm et al., 1991*; *Royer-Pokora et al., 1991*), and LMO2 has been thought to function as a bridging factor in large transcriptional complexes with several DNA-binding and adaptor proteins (TAL1/SCL, E2A, GATA1, and Ldb1) (*Wadman et al., 1997*; *Grütz et al., 1998*; *El Omari et al., 2013*; *Layer et al., 2016*). Although deletion of *Lmo2* in mice causes embryonic lethality due to embryonic erythropoiesis deficiency around 10 days post-fertilization (*Yamada et al., 1998*), conditional disruption of *Lmo2* in T-lineage committed pro-T stages results in normal T-cell development in the thymus (*McCormack et al., 2003*), indicating that LMO2 is dispensable for T-cell development after T-lineage commitment. However, it remains unclear whether LMO2 plays an important role before T-lineage commitment, including the maintenance of T-lineage potential in pre-thymic progenitors.

It is well known that the recent gene therapy trials using retrovirus-mediated introduction of the common cytokine receptor gamma chain (γc; CD132) in X-linked severe combined immunodeficiency (X-SCID) patients resulted in the development of T-cell leukemia due to retroviral insertion at the *LMO2* locus (*Hacein-Bey-Abina et al., 2003*; *Hacein-Bey-Abina et al., 2008*; *Howe et al., 2008*). Moreover, transgenic overexpression of *Lmo2* in various murine tissues only results in T-cell leukemia, indicating its involvement exclusively in T-cell malignancies when abnormally expressed (*Neale et al., 1995*). On the other hand, *Lmo2* is more highly expressed in HSCs and HPCs than in mature blood cells (*Yoshida et al., 2019*) and is known to induce reprogramming of HSCs from differentiated blood cells (*Riddell et al., 2014*) and fibroblasts (*Batta et al., 2014*; *Vereide et al., 2014*), suggesting that LMO2 actively contributes to the maintenance of the undifferentiated state of HSCs. Consistently, aberrant induction of *Lmo2* in thymocytes generates self-renewing cells that retain the capacity for T-cell differentiation (*McCormack et al., 2010*; *Cleveland et al., 2013*). Thus, it is not clear why *Lmo2* overexpression causes only T-cell malignancies and how LMO2 contributes to the maintenance of an undifferentiated state.

In attempts to establish HPC lines with T-cell differentiation potential, it has been found that pro-B cells without transcription factors essential for early B-cell development, including Pax5, Ebf1, and E2A (encoded by *Tcf3*), can be maintained in vitro with an HPC-like phenotype, retaining T-cell potential on OP9 stromal cells in the presence of IL-7 (*Nutt et al., 1999*; *Rolink et al., 1999*; *Ikawa et al., 2004*; *Pongubala et al., 2008*). In this study, based on their protocol, we established pro-B cell lines, called pro-B(+) cells, that retain T-cell potential on OP9 stromal cells. In contrast, proliferating pro-B cells, named pro-B(−) cells, were also obtained in our originally established thymic stromal cells, TD7, which die abruptly immediately following activation of Dll4-mediated Notch signaling in vitro. Thereafter, we identified *Lmo2* as a gene whose expression is downregulated in pro-B(−) cells compared to pro-B(+) cells and found that the forced expression of *Lmo2* was sufficient for the pro-B(−) cells to reacquire the ability to differentiate into T-cell lineage by Notch signaling. Furthermore, LMO2 ensures survival of pro-B cells through the activation of the Bcl11a/Bcl2 pathway and contributes to maintaining the accessibility of the *Tcf7* locus, which is one of the earliest Notch downstream targets in thymic immigrant cells. These epigenetic alterations could be mediated by direct binding of the transcriptional complex, including LMO2, to the target loci. Together with loss-of-function experiments, we demonstrate that LMO2 is significant for maintaining T-cell progenitors for the progression of the T-cell differentiation program initiated by Notch signaling.

## Results

### LMO2 has a crucial role in the maintenance of T-cell differentiation capacity in *Ebf1*-deficient pro-B cells

Using different stromal cells, OP9 and TD7, with fetal liver progenitor cells from *Ebf1*-deficient mice, we established two types of pro-B cell lines, with and without the ability to differentiate into T-cell lineages designated pro-B(+) and pro-B(−) cells, respectively (*Figure 1—figure supplement 1A,B*). Both the *Ebf1*-deficient pro-B lines grew robustly on OP9 cells with IL-7, Flt3L, and SCF (*Figure 1A*, upper panels) and expressed intermediate levels of B220 (*Figure 1—figure supplement 1C*). Pro-B (+) cells were able to initiate differentiation into the T-cell lineage, DN2 (CD25$^+$CD44$^+$), DN3 (CD25$^+$CD44$^{lo}$), and double-positive (DP) stages on Dll4-expressing OP9 (OP9-Dll4) cells in vitro (*Figure 1A*, lower right, *Figure 1—figure supplement 1D*), and mature into CD4 and CD8 T cells in vivo (*Figure 1—figure supplement 1E*), as shown previously (*Pongubala et al., 2008*). In contrast, pro-B(−) cells remained in the pro-B cell phenotype (CD25$^-$CD44$^+$B220$^{int}$) and died with Dll4-mediated Notch stimulation, although the expression levels of the Notch receptors were comparable (*Figure 1A*, *Figure 1—figure supplement 1F*). To explore their different characteristics at the molecular level, we carried out comparative microarray analysis and identified the differentially expressed genes between pro-B(+) and pro-B(−) cells (*Figure 1B*, *Supplementary file 1*). We identified approximately 400 differentially expressed genes (FC > 10) between the pro-B(+) and pro-B(−) cells, and these genes were enriched for genes related to the 'Hematopoietic cell lineage' pathway (*Figure 1—figure supplement 1G*). Among them, the functional importance of *Meis1*, *Hmga2*, and *Bcl11a* has been reported in undifferentiated HPCs (*Wong et al., 2007*; *Ariki et al., 2014*; *Nishino et al., 2008*; *Copley et al., 2013*; *Yu et al., 2012*). However, the introduction of *Meis1* and *Hmga2* failed to overcome the defective T-cell differentiation in the presence of Notch signaling, and that of *Bcl11a* was highly toxic in pro-B(−) cells (data not shown). Lmo2 has also been shown to be linked to the reprogramming of lineage-committed blood cells or mesenchymal cells to the induced HSCs (*Riddell et al., 2014*; *Batta et al., 2014*; *Vereide et al., 2014*). We found that the expression levels of *Lmo2* mRNA and protein were approximately threefold higher in pro-B(+) than pro-B(−) cells (*Figure 1B,C*), and enforced expression of *Lmo2* markedly provided their capacity to differentiate into T-cell lineage following Notch stimulation in vitro (*Figure 1D*). Therefore, these results suggest that LMO2 plays a pivotal role in the maintenance of the capacity to differentiate into T-cell lineage driven by Notch signaling in *Ebf1*-deficient pro-B cells.

### Disruption of *Lmo2* in pro-B(+) cells leads to differentiation arrest following stimulation by Notch signaling

To confirm the necessity of LMO2 for the maintenance of the differentiation capacity into T-cell lineage, we tested the effect of *Lmo2* disruption on the developmental potential of T-cell lineage in bone marrow (BM)-derived progenitor cells and pro-B(+) cells. BM progenitors from Cas9;Bcl2 Tg mice were infected with a bicistronic retroviral vector carrying an sgRNA against *Lmo2* with the human nerve growth factor receptor (hNGFR) or control sgRNA against luciferase. Two days after sgRNA introduction, the cells were transferred onto an OP9-Dll1 monolayer to initiate T-lineage differentiation (*Figure 2—figure supplement 1A*). The ability sgLMO2-treated progenitors to differentiate into the T-lineage was comparable to that of control (*Figure 2—figure supplement 1B*). Next, Cas9-introduced pro-B(+) cells were transduced with sgRNA against LMO2, which caused specific loss of LMO2 protein 4 days after retroviral infection as detected by immunoblotting (*Figure 2—figure supplement 1C*). Five or ten days after sgLMO2 transduction, the cells were transferred onto a OP9-Dll4 monolayer to induce Notch-dependent T-lineage differentiation (*Figure 2A*). *Lmo2*-deficient pro-B(+) cells progressed to the DN2 stage (CD25$^+$CD44$^+$) as well as control cells on day 5, however, developmental arrest was observed ten days after sgLMO2 transduction (*Figure 2B*). These results indicate that LMO2 is necessary for pro-B(+) cells to maintain their ability to differentiate into T-cell lineage, and loss of their differentiation potential took several days (~10 days) after *Lmo2* disruption. Thus, the process may require loss of LMO2-mediated slower time-scale transcriptional changes, including histone modifications, chromatin remodeling, and DNA methylation.

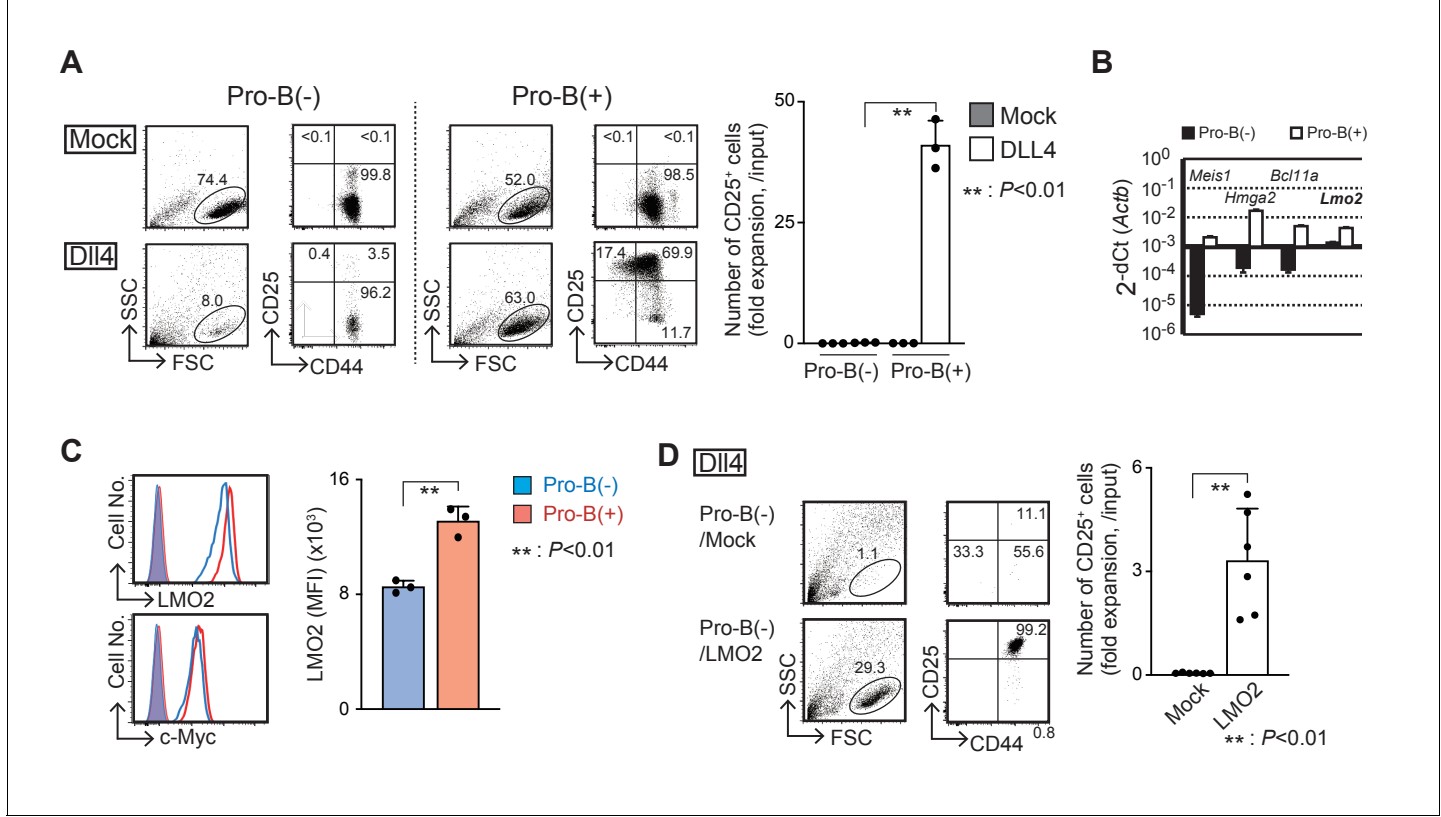

**Figure 1.** LMO2 is critical for the maintenance of T-cell differentiation potential in *Ebf1*-deficient pro-B cells. (A) Establishment of *Ebf1*-deficient pro-B cell lines with or without differentiation potential to the T-cell lineage. Lineage markers (CD19, Gr1, TER119, NK1.1)-negative, c-kit-positive cells in *Ebf1*$^{-/-}$ FL were cultured on TD7 or OP9 cells, and *Ebf1*-deficient pro-B cell lines were established. Stably growing pro-B cells with or without T-cell potential (pro-B(+) or pro-B(−)) were cultured on OP9-Mock (Mock) or OP9-Dll4 (Dll4) cells with Flt3L, SCF, and IL7 for 6 days and analyzed for the expression of CD44 and CD25 (right panels) in the lymphoid cell gate (FSC vs. SSC, left panels) by flow cytometry. The numbers in the profiles indicate the relative percentages in each corresponding quadrant or fraction. Numbers of CD25$^+$ cells (fold expansion/input) are shown with standard deviation (SD) (right). Statistical analysis was performed using the two-tailed Student's t-test. **p<0.01. Data are representative of three independent experiments with similar results. (B) Reverse transcription (RT)-quantitative PCR (qPCR) detection of *Meis1*, *Hmga2*, *Bcl11a*, or *Lmo2* transcripts in pro-B(−) (closed columns) and pro-B(+) (open columns) cells. Data represent the mean values of three independent biological replicates, and all values are normalized to the expression of *Actb*. Error bars indicate SD. Three independent experiments were performed, and similar results were obtained. (C) Representative intracellular staining profiles of LMO2 and c-Myc in pro-B(−) (open blue line) and pro-B(+) (open red line) cells are shown. Closed lines (orange) represent staining with control rabbit mAb of pro-B(−) and pro-B(+), which were completely merged. The average mean fluorescent intensity (MFI) of LMO2 is shown with SD (right). A two-tailed Student's t-test was used for statistical analysis. **p<0.01. Three independent experiments were performed with similar results. (D) Introduction of *Lmo2* is sufficient to maintain the T-cell differentiation potential in pro-B cells. Empty vector- or *Lmo2*-transduced pro-B(−) cells (pro-B(−)/Mock or pro-B(−)/LMO2) were cultured on OP9-Dll4 for 6 days and analyzed for the expression of CD44 and CD25 (right panels) in lymphoid cell gate (left panels) and rat CD2$^+$ (lentivirus-infected) CD45$^+$ fraction. Numbers of CD25$^+$ cells (relative expansion/input) are shown with SD (right). **p<0.01 by two-sided Student's t-test. Six independent experiments were performed with similar results.

The online version of this article includes the following source data and figure supplement(s) for figure 1:

**Source data 1.** Raw data used to generate the graph in *Figure 1A*.
**Source data 2.** Raw data used to generate the graph in *Figure 1B*.
**Source data 3.** Raw data used to generate the graph in *Figure 1C*.
**Source data 4.** Raw data used to generate the graph in *Figure 1D*.
**Figure supplement 1.** Characterization of pro-B(+) and pro-B(−) cells.

## LMO2 regulates survival of *Ebf1*-deficient pro-B cells via Bcl11a/Bcl2 pathway

To identify the downstream targets of LMO2, we compared the gene expression profiles of pro-B (−), *Lmo2*-introduced pro-B(−) (pro-B(−)/LMO2), *Meis1*-introduced pro-B(−) (pro-B(−)/Meis1), and pro-B(+) cells (*Figure 3A*). We found that *Lmo2*-introduction restored the expression of *Bcl11a* in

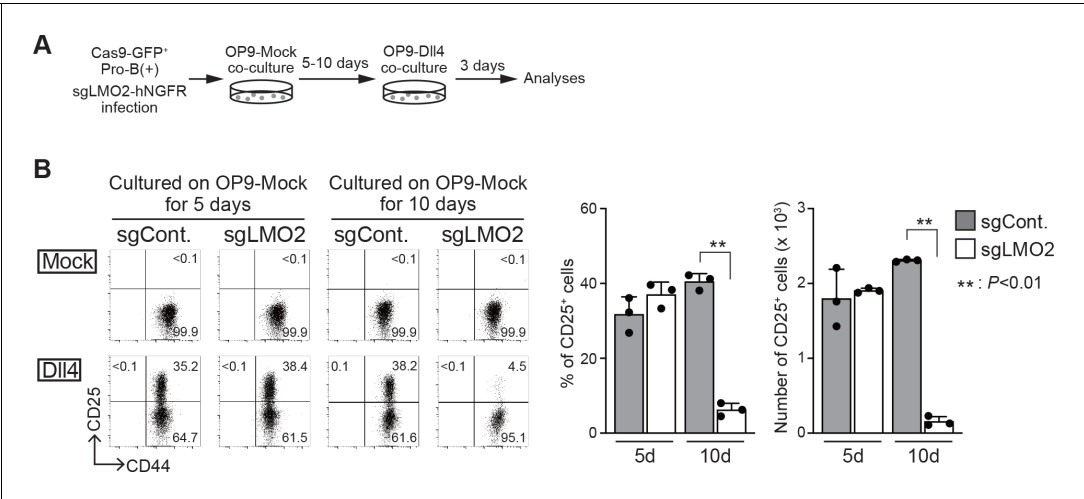

**Figure 2.** Loss of *Lmo2* leads to the differentiation arrest in pro-B(+) cells. (**A**) An experimental scheme for the deletion of *Lmo2* using the CRISPR/Cas9 system in pro-B cell lines is shown. (**B**) Retroviral vectors encoding sgRNA against luciferase (sgCont.) or LMO2 (sgLMO2) were introduced into Cas9-expressing (GFP⁺) pro-B(+) cells. Five (cultured on OP9-Mock for 5 days, left panels) or 10 days (cultured on OP9-Mock for 10 days, right panels) after co-cultured on OP9-Mock cells following the infection, pro-B cells were cultured again on OP9-Mock (Mock) or OP9-Dll4 (Dll4) stromal cells for 3 days. GFP⁺hNGFR⁺ sgRNA-transduced cells were gated and analyzed for CD44 and CD25 expression (left). The percentages and numbers of CD25⁺ cells among GFP⁺hNGFR⁺ sgRNA-transduced cells, cultured on OP9-Dll4, are shown with SD (right). The data represent the mean values of three independent biological replicates. Each value is indicated by a closed circle. \*\*p<0.01 by two-sided Student's t-test.

The online version of this article includes the following source data and figure supplement(s) for figure 2:

**Source data 1.** Raw data used to generate the graph in *Figure 2B*.
**Figure supplement 1.** CRISPR/Cas9-mediated deletion of *Lmo2* in BM progenitors.
**Figure supplement 1—source data 1.** Original data used to generate the panels in *Figure 2—figure supplement 1C*.
**Figure supplement 1—source data 2.** Raw data used to generate the graph in *Figure 2—figure supplement 1B*.

pro-B(−) cells, whereas *Meis1* further downregulated *Bcl11a* expression. The expression of other potential target genes, including *Meis1* and *Hmga2*, were also changed following *Lmo2-* and *Meis1-* transduction, which failed to restore T-lineage differentiation capacity in pro-B(−) cells (*Figure 3A*). Bcl11a is known to regulate the survival of lymphoid progenitors via induction of the anti-apoptotic gene, *Bcl2* (*Yu et al., 2012*). Therefore, we examined the roles of the Bcl11a/Bcl2 pathway in the survival of pro-B(−) cells and their potential to differentiate into T-lineage cells. Transduction of *Bcl2* significantly protected pro-B(−) cells from cell death, before and after Notch stimulation (*Figure 3B*, *Figure 3—figure supplement 1*). Conversely, overexpression of the oncogene, *Bcl11a,* was highly toxic in pro-B cells. Importantly, *Bcl2*-introduced pro-B(−) cells (pro-B(−)/Bcl2) still failed to progress into the CD25⁺ DN2 stage (*Figure 3C*). Thus, these results suggest that LMO2 would regulate survival of pro-B(−) cells via the Bcl11a/Bcl2 pathway, and other mechanisms contribute to the maintenance of the potential to differentiate into T-cell lineage.

## LMO2 is required for the activation of *Tcf7* after Notch signaling

Our comparative microarray analysis of pro-B(−) and pro-B(+) cells showed that expression of *Tcf7* (encoding TCF1), one of the most important and the earliest Notch target genes in thymic immigrant cells, was weakly detected in pro-B(+) cells; however, its level was approximately 100-fold lower in pro-B(−) cells. Therefore, we next examined the expression kinetics of *Tcf7* in pro-B(+)/Bcl2 and pro-B(−)/Bcl2 cells that survive after Notch stimulation but fail to differentiate into T-cell lineages. The upregulation of *Tcf7* expression observed in pro-B(+)/Bcl2 cells after Notch stimulation was significantly abrogated in pro-B(−)/Bcl2 cells (*Figure 4A*). In contrast, another Notch-regulated gene, *Gata3*, was upregulated by Notch stimulation but modestly lower in pro-B(−)/Bcl2 cells compared to that in pro-B(+)/Bcl2 cells (*Figure 4A*). These results were further confirmed at the protein level using pro-B(−)/Bcl2 and pro-B(+)/Bcl2 cells at 7 days post-Notch stimulation (*Figure 4B*). In addition, there was a decrease in *Tcf7* expression in *Lmo2*-deficient pro-B(+) cells (*Figure 2*) after 10 days but

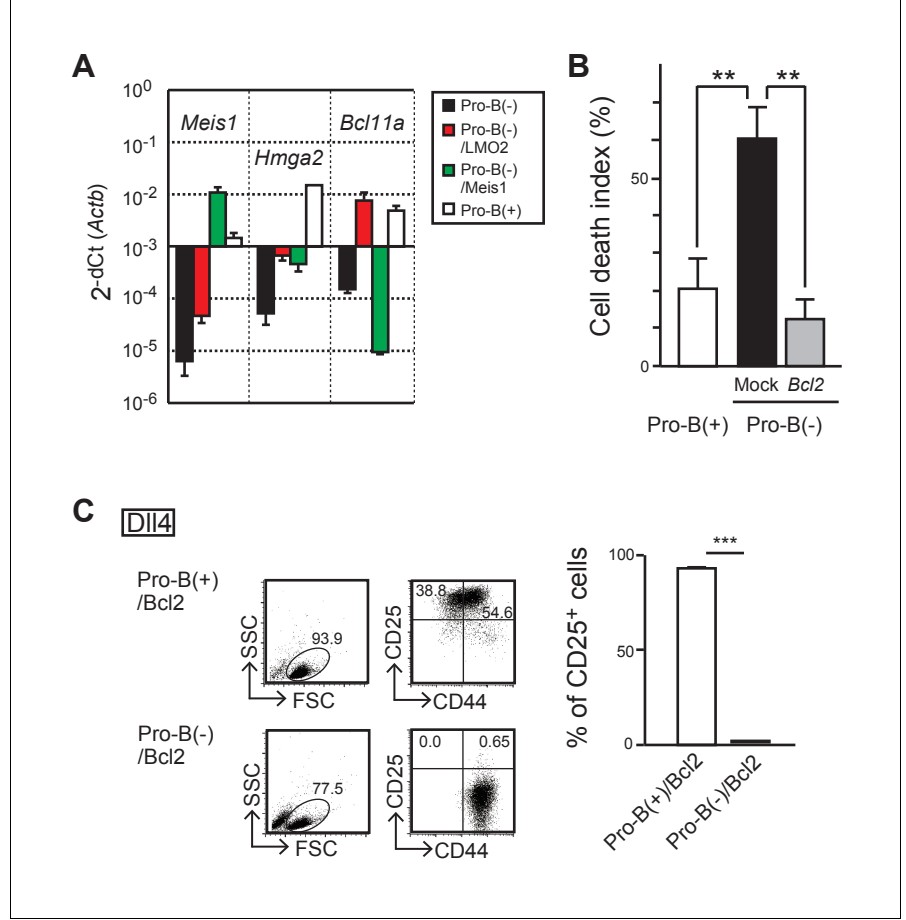

**Figure 3.** LMO2 regulates survival of *Ebf1*-deficient pro-B cells via Bcl11a/Bcl2 pathway. (**A**) RT-qPCR detection of *Meis1*, *Hmga2*, and *Bcl11a* transcripts in *Lmo2*- (red column), *Meis1*-transfected (green column) pro-B(−) cells, parent pro-B(−) (black column), and pro-B(+) (white column) cells, as shown in *Figure 1B*. Two independent experiments were performed with similar results. (**B**) Overexpression of *Bcl2* improves cell survival of pro-B(−) cells. Pro-B(+) cells (open column) and empty vector (mock) or human *BCL2* (*Bcl2*)-transduced pro-B(−) cells were cultured on OP9-Mock or OP9-Dll4 for 2 days. After culturing, the dead cells were detected by staining for Annexin V and 7-AAD in CD45$^+$ and hNGFR$^+$ (retrovirus-infected) cell populations (*Figure 3—figure supplement 1*). The cell death index was calculated as the difference in the percentage of dead cells in pro-B cells after co-culturing with OP9-Mock and OP9-Dll4. The data represent the mean values of three independent biological replicates with SD. \*\*p<0.01 by two-sided Student's t-test. (**C**) Bcl2 overexpression does not provide differentiation potential in pro-B(−) cells. Pro-B(−) or pro-B(+) cells with human *BCL2* (pro-B(−)/Bcl2, pro-B(+)/Bcl2) were cultured on OP9-Dll4 as shown in *Figure 1D*. After culturing, the live cells were analyzed for the expression of CD44 and CD25 (left). The data represent the mean values of percentages of CD25$^+$ cells in three independent biological replicates with SD (right). \*\*\*p<0.001 by two-sided Student's t-test.

The online version of this article includes the following source data and figure supplement(s) for figure 3:

**Source data 1.** Raw data used to generate the graph in *Figure 3A*.
**Source data 2.** Raw data used to generate the graph in *Figure 3B*.
**Source data 3.** Raw data used to generate the graph in *Figure 3C*.
**Figure supplement 1.** Overexpression of *Bcl2* improves the cell survival in pro-B(−) cells.
**Figure supplement 1—source data 1.** Raw data used to generate the graph in *Figure 3—figure supplement 1*.

not after 5 days of sgRNA transduction (*Figure 4—figure supplement 1*). We then tested the role of *Tcf7* using a gain-of-function strategy and found that introduction of *Tcf7* clearly rescued the defect in the differentiation capacity of pro-B(−)/Bcl2 into T-cell lineage (DN2 stage), after Notch signaling was provided (*Figure 4C*). These results indicate that LMO2 expression in T-cell

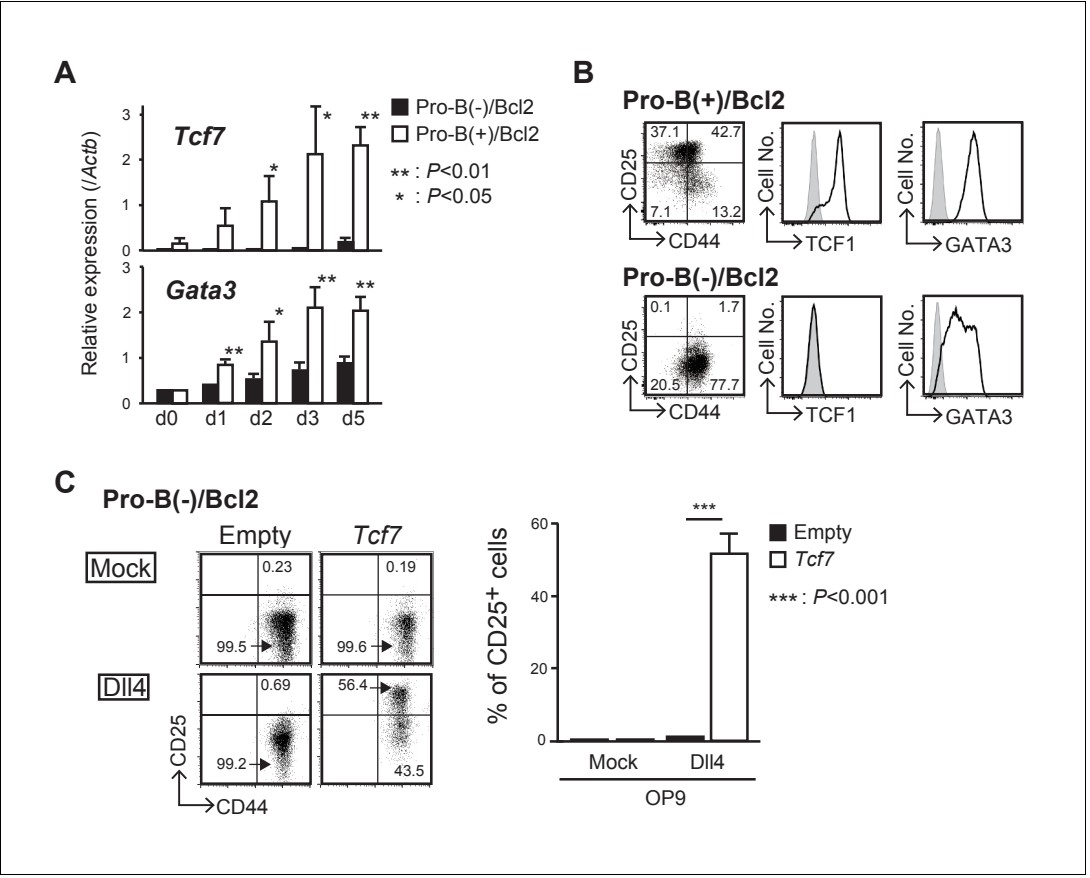

**Figure 4.** LMO2 is required for activation of *Tcf7* after Notch signaling. (**A**) Expression levels of *Tcf7* and *Gata3* in pro-B(−) and pro-B(+) cells with exogenous BCL2, at 0–5 days after the culture on OP9-Dll4, were analyzed by RT-qPCR. The relative expression (/*Actb*) is shown with SD. *p<0.05, **p<0.01 by two-sided Student's t-test. (**B**) Intracellular staining of TCF1 or GATA3 in pro-B(+)/Bcl2 (upper panels) and pro-B(−)/Bcl2 (lower panels) was performed at day 7 after Notch stimulation; representative expression profiles of CD44 and CD25 are also shown (left). Results are representative of three independent experiments. (**C**) Introduction of *Tcf7* provides differentiation potential for T-cell lineage to pro-B(−)/Bcl2. Pro-B(−)/Bcl2 cells were infected with either empty control or *Tcf7*-containing lentivirus, and the cells were co-cultured on OP9-Mock (upper panels) or OP9-Dll4 (lower panels) for 3 days. Lentivirus-infected cells were analyzed for the expression of CD44 and CD25. Three independent experiments were performed with similar results. The percentages of CD25$^+$ cells are shown with SD (right). ***p<0.001 by two-sided Student's t-test.

The online version of this article includes the following source data and figure supplement(s) for figure 4:

**Source data 1.** Raw data used to generate the graph in *Figure 4A*.

**Source data 2.** Raw data used to generate the graph in *Figure 4C*.

**Figure supplement 1.** Down-regulation of *Tcf7* in *Lmo2*-deficient pro-B(+) cells.

**Figure supplement 1—source data 1.** Raw data used to generate the graph in *Figure 4—figure supplement 1*.

progenitors plays a crucial role in the activation of *Tcf7* once the progenitors migrate into the thymus and are stimulated by Notch signaling.

## DNA methylation status of the *Tcf7* locus is maintained by LMO2

Next, we examined the epigenetic status of the *Tcf7* locus in pro-B cell lines. We observed that in pro-B(+) cells, the transcriptional start site (TSS) of the *Tcf7* locus was highly enriched for the active histone mark H3K4-3Me. However, *Lmo2* disruption caused a significant reduction in the H3K4-3Me marks 10 days post-sgRNA introduction (~30%), and a modest reduction in the H3K4-3Me levels after 5 days of *Lmo2* deletion (~70%) (*Figure 5—figure supplement 1*). Moreover, we found a CpG island at the TSSs of *Tcf7* (*Figure 5A*). Thus, we examined the DNA methylation levels of the CpG

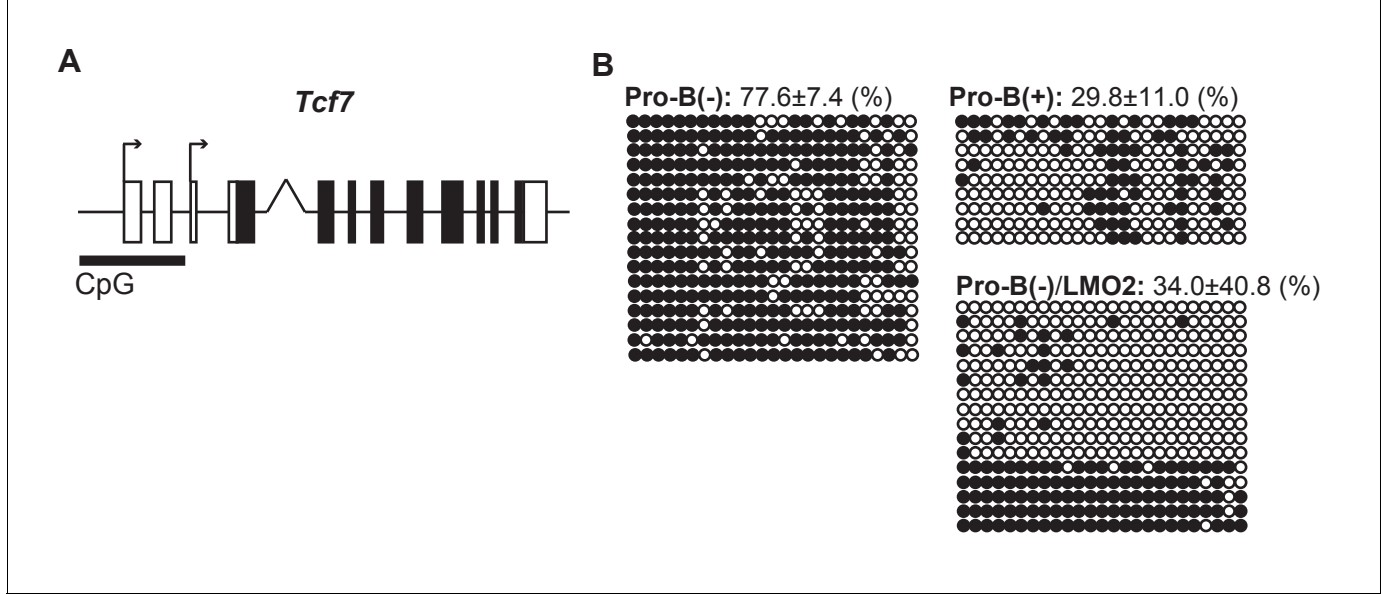

**Figure 5.** DNA methylation status of the *Tcf7* locus is maintained by LMO2. (**A**) A CpG island at the transcriptional start sites (TSSs) of the *Tcf7* locus, which contains 25 potential CpG methylation sites. (**B**) The DNA methylation status of CpG island at the TSSs of the *Tcf7* locus was determined by bisulfite sequencing in pro-B(−), pro-B(+), and LMO2-transduced pro-B(−) cells (pro-B(−)/LMO2). Bisulfite-converted genomic DNA around the TSSs of *Tcf7* was amplified using PCR, and each PCR product was sequenced. The 25 horizontal circles each represent a CpG sequence derived from a single PCR product (17 clones from pro-B(−) cells, 9 clones from pro-B(+) cells, and 16 clones from pro-B(−)/LMO2 cells). Closed and open circles indicate methylated and demethylated CpG sites, respectively. The frequencies of the methylated CpGs are shown with SD. Data are based on two independent pooled experiments.

The online version of this article includes the following source data and figure supplement(s) for figure 5:

**Figure supplement 1.** H3K4-3Me levels around the *Tcf7* locus in *Lmo2*-deficient cells.

**Figure supplement 1—source data 1.** Raw data used to generate the graph in *Figure 5—figure supplement 1*.

island at the TSSs of the *Tcf7* locus in the pro-B cell lines. In pro-B(−) cells, the CpG island at the *Tcf7* locus was highly methylated, in contrast to pro-B(+) cells that had a highly demethylated TSSs (77.6% vs. 29.8%) (*Figure 5B*). Moreover, enforced expression of *Lmo2* in pro-B(−) cells induced demethylation of the CpG island at the TSSs (*Figure 5B*, pro-B(−)/LMO2). These results demonstrate that the epigenetic status of the *Tcf7* locus in the progenitor cells is maintained by LMO2 in a transcriptionally poised chromatin state for quick responsiveness following Notch stimulation.

## LMO2 directly binds to the *Bcl11a* and *Tcf7* loci

Finally, we performed chromatin immunoprecipitation followed by deep-sequencing (ChIP-seq) analysis to identify genome-wide LMO2 occupancy sites in the pro-B(+) cells. We identified more than 1500 reproducible LMO2 binding peaks, and enrichment of Runx and ETS motifs that are frequently found in open chromatin regions of hematopoietic progenitors and pre-commitment pro-T cells (*Yoshida et al., 2019*; *Ungerbäck et al., 2018*; *Figure 6A*). The motif for a previously reported LMO2 interacting partner, the bHLH factors, was also coenriched (*Figure 6A*). Many of the bHLH-regulated genes, including *Lyl1*, *Erg*, and *Hhex*, were bound by LMO2 (*Figure 6B*). In addition, these motifs are highly relevant to the binding sites for major transcription factors, such as LMO2, in HPCs (*Wilson et al., 2010*). Moreover, two LMO2 binding peaks were detected in the downstream regions of the *Bcl11a* locus, one of the LMO2-sensitive genes (*Figures 3A* and *6B*). Most importantly, a clear LMO2 peak was observed at the −35 kb upstream region of the *Tcf7* locus. This binding site overlapped with RBPJ, a DNA-binding subunit of the Notch intracellular domain (NICD) transcriptional complex, binding sites at the *Tcf7* locus, and co-occupied with Runx1 in pre-commitment DN1 cells (GSE148441, GSE110020) (*Romero-Wolf et al., 2020*; *Hosokawa et al., 2018*; *Shin et al., 2021*; *Figure 6B*, *Figure 6—figure supplement 1*). Taken together, LMO2 directly binds to the *Bcl11a* and

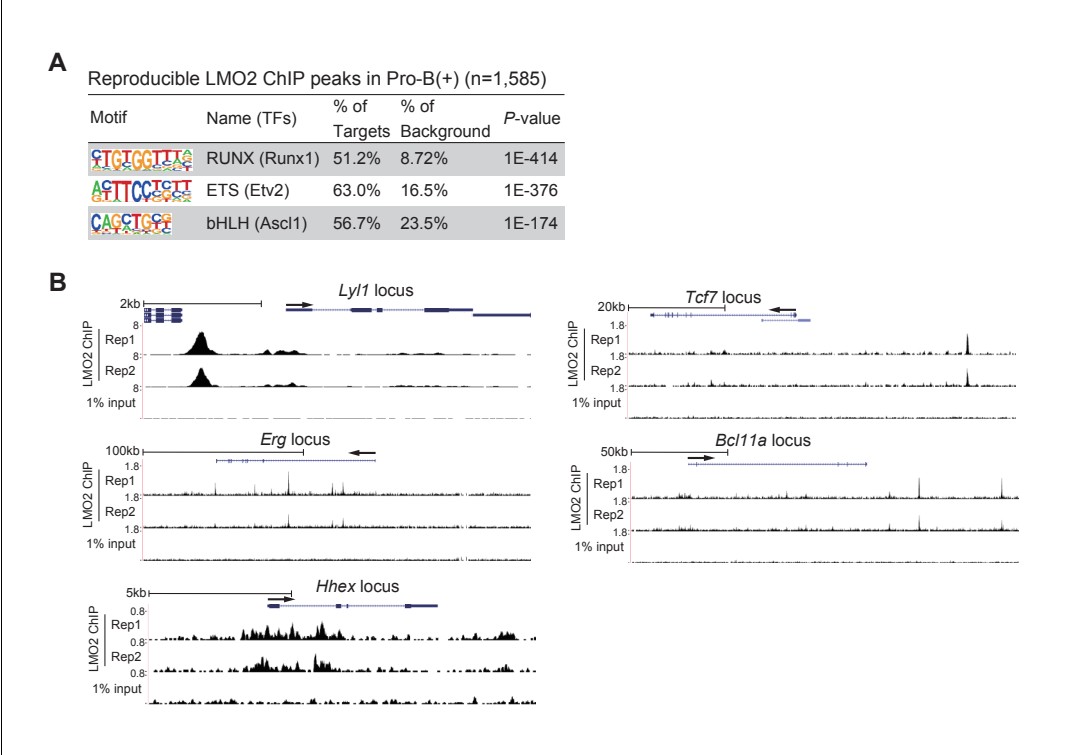

**Figure 6.** LMO2 binds to the upstream region of the *Tcf7* locus. (**A**) LMO2 ChIP-seq analyses were performed using the pro-B(+) cell line. The top three enriched sequence motifs among the 1585 reproducible LMO2 peaks are shown. Data are based on ChIP-seq peaks scored as reproducible in two replicate samples. (**B**) ChIP-seq tracks showing two replicates of LMO2 binding profiles around the *Lyl1*, *Erg*, *Hhex*, *Tcf7*, and *Bcl11a* loci in pro-B(+) cell line with tracks for 1% input.

The online version of this article includes the following figure supplement(s) for figure 6:

**Figure supplement 1.** LMO2 binds to one of the RBPJ and Runx1 binding sites at the *Tcf7* locus.

*Tcf7* loci and regulates their expression to maintain the ability of progenitors to differentiate into T-cell lineage.

## Discussion

Here, we demonstrate that LMO2 contributes to the differentiation capacity of T-cell lineage in *Ebf1*-deficient pro-B cells via activation of the Bcl11a/Bcl2 pathway, which is critical for cell survival, and maintenance of DNA methylation status of the *Tcf7* locus. LMO2 directly binds to its target loci and enables the *Tcf7* locus to achieve a state that is responsive to T-cell induction following Notch signaling.

We established our original thymic stromal cell line, TD7, in which HSCs differentiate into CD19[+] B-lineage cells in the presence of IL-7 (*Figure 1—figure supplement 1A*) or into Thy1[+]CD25[+] T-lineage cells upon receiving Notch signaling. To establish pro-B cell lines possessing pluripotency, we cultured HSCs from the fetal liver of *Ebf1*-deficient mice on TD7 or OP9 stromal cells, which have been reported to establish pro-B cell lines (*Pongubala et al., 2008*). The pro-B cells derived from cultures with TD7 grew robustly in the presence of IL-7, but abruptly died when Notch signaling is provided, in contrast to the cells from cultures with OP9, which differentiated into DN2/3 stages with Notch signaling. The characteristics observed in the former pro-B cells were found to be due to decreased expression of *Lmo2*. These results suggest that TD7 does not support the pluripotency of *Ebf1*-deficient pro-B cells, leading to loss of LMO2. Although the molecular machinery underlying

LMO2 expression in HSCs has not been fully described, it was shown that several cis-regulatory elements around the *Lmo2* locus cooperatively function and ensure the full expression pattern of *Lmo2* (*Landry et al., 2005*; *Landry et al., 2009*), which appears to be driven in part by PU.1, TAL1, GATA factors, and LMO2 itself. The culture conditions provided by TD7 may be unable to sustain sufficient expression of these transcription factors. Alternatively, it is possible that TD7 could not support pro-B cells that retained their pluripotency because M-CSF, which is not produced by OP9 (*Nakano et al., 1994*), promoted the differentiation of HSCs into myeloid cells. In fact, the establishment of pro-B cells on TD7 was accompanied by a prolonged emergence of myeloid marker-positive cells compared to that on OP9, and the remaining pro-B cells lost their pluripotency (*Figure 1—figure supplement 1B*). Therefore, HSCs may be depleted by differentiation into the myeloid lineage, resulting in IL-7-dependent proliferation of pro-B cells without pluripotency that express reduced *Lmo2*. In either case, substantial expression of *Lmo2* is critical for the maintenance of pluripotency in *Ebf1*-deficient pro-B cells, which is consistent with the finding that LMO2 is an important factor for the reprogramming of committed blood or mesenchymal cells to the induced HSCs (*Riddell et al., 2014*; *Batta et al., 2014*; *Vereide et al., 2014*).

A remarkable feature of LMO2 is that it only generates T-cell malignancies by abnormal expression (*Neale et al., 1995*; *Curtis and McCormack, 2010*; *Matthews et al., 2013*). A recent study indicated that LMO2 overexpression induces T-cell malignancies not only at the hematopoietic undifferentiated stages, but also after differentiation to B-lineage cells (*García-Ramírez et al., 2018*). However, the strong association between LMO2 and T-cell malignancy is not well understood at the molecular level. We showed here that LMO2 plays an important role in maintaining the chromatin structure of the *Tcf7* locus in an accessible state necessary for Notch signaling-mediated activation of *Tcf7*. The expression of *Tcf7* is highly specific to the T-cell lineage among hematopoietic cells, and TCF1 is important for initiating the epigenetic identity of the T-cell lineage (*Johnson et al., 2018*). Therefore, it can be inferred that, as the *Tcf7* locus remains in an accessible state in HPCs or lineage-committed cells with ectopic LMO2 expression, they efficiently differentiate into T-lineage cells and selectively develop T-cell malignancy. In addition, TCF1 forms a complex with β-catenin, which contributes to the development of T-cell malignancy (*Bigas et al., 2020*). Although the contribution of the β-catenin/TCF1 complex in normal T-cell development remains unclear, the complex, together with RAG molecules, induces various genetic instabilities including those in the *Myc* gene in immature T cells and causes *Myc* gene-targeted tumorigenesis (*Dose et al., 2014*; *Gekas et al., 2016*). As TCF1 is also involved in other tumorigenesis (*Liu et al., 2019*), aberrant expression of LMO2 may promote T-cell malignancy via TCF1.

Previous reports suggest that LMO2 forms complexes with the bHLH transcription factors, including E2A, Lyl1, and TAL1/SCL, and binds to their target genomic sites (*Wadman et al., 1997*; *Grütz et al., 1998*; *El Omari et al., 2013*; *Layer et al., 2016*). Consistently, in this study, we found that the binding motif for bHLH factors was enriched in LMO2 binding regions in pro-B(+) cells. Thus, LMO2 seems to function along with these bHLH factors in the lymphoid progenitors. In addition, we found that LMO2 directly binds to the *Tcf7* locus. Initiation of *Tcf7* expression requires Notch signaling and Runx factors in T-cell progenitors (*Weber et al., 2011*; *Guo et al., 2008*; *Shin et al., 2021*). Two NICD-RBPJ complex and Runx1 binding sites were found at the −31 and −35 kb upstream regions of the *Tcf7* locus (*Figure 6—figure supplement 1*; *Weber et al., 2011*; *Romero-Wolf et al., 2020*; *Hosokawa et al., 2018*). Among these, the functional importance of the −31 kb region has been previously reported as a Notch-dependent enhancer of *Tcf7* (*Weber et al., 2011*; *Harly et al., 2020*). The LMO2 binding site, identified in this study, overlapped with the other RBPJ binding site, at the −35 kb upstream region. While the physiological role of the −35 kb RBPJ binding site has not been clarified, our data suggest that direct binding of LMO2 to this region in progenitor cells plays an important role in maintaining the accessible chromatin configuration of the *Tcf7* locus. In fact, *Lmo2* expression levels were found to be associated with DNA methylation status around the *Tcf7* locus and subsequent Notch-mediated activation of *Tcf7* expression. Taken together, in the T-cell progenitor stage, LMO2 acts as a gatekeeper for maintaining the transcriptionally poised chromatin state of the *Tcf7* locus via direct binding to the −35 kb region, and guarantees T-cell differentiation potential.

# Materials and methods

**Key resources table**

| Reagent type (species) or resource | Designation | Source or reference | Identifiers | Additional information |
|---|---|---|---|---|
| Genetic reagent (*Mus musculus*) | *Ebf1*$^{+/-}$ | *Pongubala et al., 2008* | | Provided by Dr. Grosschedl, Max Planck Institute of Immunobiology and Epigenetics |
| Genetic reagent (*Mus musculus*) | B6.Cg-Tg(BCL2)25Wehi/J | Jackson Laboratory | Stock# 002320 | |
| Genetic reagent (*Mus musculus*) | B6.Gt(ROSA) 26$^{Sortm1.1(CAG-cas9*, -EGFP)Fezh}$/J | Jackson Laboratory | Stock# 024858 | |
| Cell line (*Mus musculus*) | OP9 | *Yokoyama et al., 2013* | Stromal cell line derived from fetal murine calvaria (B6 x C3H, *op/op*) | |
| Cell line (*Mus musculus*) | TD7 | This paper | | Mouse fetal thymus (B6, E15.5)-derived mesenchymal cell line |
| Cell line (*Mus musculus*) | Pro-B(+) | This paper | | Ebf1-deficient fetal liver-derived hematopoietic progenitor cell line |
| Cell line (*Mus musculus*) | Pro-B(−) | This paper | | Ebf1-deficient fetal liver-derived hematopoietic progenitor cell line |
| Cell line (*Mus musculus*) | OP9-Dll4 | *Hirano et al., 2020* | | |
| Cell line (*Homo sapiens*) | HEK293T | *Hirano et al., 2020* | RRID:CVCL_0063 | |
| Cell line (*Homo sapiens*) | PLAT-E | *Hirano et al., 2020* | RRID:CVCL_B488 | |
| Antibody | FITC anti-mouse CD4 (Rat monoclonal) | BD Biosciences | Cat# 561835 RRID:AB_10894386 | FC (1:500) |
| Antibody | PE anti-mouse CD4 (Rat monoclonal) | BD Biosciences | Cat# 561829 RRID:AB_10926205 | FC (1:500) |
| Antibody | APC anti-mouse CD8a (Rat monoclonal) | BioLegend | Cat# 100711 RRID:AB_312750 | FC (1:500) |
| Antibody | APCCy7 anti-mouse CD8a (Rat monoclonal) | BioLegend | Cat# 100713 RRID:AB_312752 | FC (1:500) |
| Antibody | PerCpCy5.5 anti-mouse CD11b (Rat monoclonal) | BioLegend | Cat# 101227 RRID:AB_893233 | FC (1:500) |
| Antibody | PECy7 anti-mouse CD19 (Rat monoclonal) | Tonbo Biosciences | Cat# 60-0193 RRID:AB_2621840 | FC (1:250) |
| Antibody | PerCpCy5.5 anti-mouse CD25 (Rat monoclonal) | eBioscience | Cat# 45-0251-82 RRID:AB_914324 | FC (1:1000) |
| Antibody | APC-e780 anti-mouse CD25 (Rat monoclonal) | eBioscience | Cat# 47-0251-82 RRID:AB_1272179 | FC (1:200) |
| Antibody | FITC anti-mouse CD44 (Rat monoclonal) | BioLegend | Cat# 103005 RRID:AB_312956 | FC (1:500) |
| Antibody | APCCy7 anti-mouse CD44 (Rat monoclonal) | BioLegend | Cat# 103027 RRID:AB_830784 | FC (1:500) |
| Antibody | PECy7 anti-mouse CD45 (Rat monoclonal) | eBioscience | Cat# 25-0451-82 RRID:AB_2734986 | FC (1:400) |

*Continued on next page*

*Continued*

| Reagent type (species) or resource | Designation | Source or reference | Identifiers | Additional information |
|---|---|---|---|---|
| Antibody | APC anti-mouse CD45 (Rat monoclonal) | BioLegend | Cat# 103111 RRID:AB_312976 | FC (1:1000) |
| Antibody | PE anti-mouse CD45.1 (Mouse monoclonal) | BioLegend | Cat# 110727 RRID:AB_893348 | FC (1:250) |
| Antibody | PerCpCy5.5 anti-mouse CD45.2 (Mouse monoclonal) | BioLegend | Cat# 109807 RRID:AB_313444 | FC (1:250) |
| Antibody | PerCpCy5.5 anti-mouse B220 (Rat monoclonal) | BioLegend | Cat# 103235 RRID:AB_893356 | FC (1:200) |
| Antibody | APCCy7 anti-mouse Thy1.2 (Rat monoclonal) | BioLegend | Cat# 105327 RRID:AB_10613280 | FC (1:1000) |
| Antibody | FITC anti-mouse Gr-1 (Rat monoclonal) | BioLegend | Cat# 108405 RRID:AB_313370 | FC (1:500) |
| Antibody | Biotin anti-mouse Gr-1 (Rat monoclonal) | eBioscience | Cat# 13-5931-86 RRID:AB_466802 | FC (1:300) |
| Antibody | Biotin anti-mouse TER119 (Rat monoclonal) | eBioscience | Cat# 13-5921-85 RRID:AB_466798 | FC (1:300) |
| Antibody | Biotin anti-mouse CD11b (Rat monoclonal) | eBioscience | Cat# 13-0112-86 RRID:AB_466361 | FC (1:300) |
| Antibody | Biotin anti-mouse CD11c (Armenian Hamster monoclonal) | eBioscience | Cat# 13-0114-85 RRID:AB_466364 | FC (1:300) |
| Antibody | Biotin anti-mouse CD19 (Rat monoclonal) | eBioscience | Cat# 13-0193-85 RRID:AB_657658 | FC (1:300) |
| Antibody | Biotin anti-mouse NK1.1 (Rat monoclonal) | eBioscience | Cat# 13-5941-85 RRID:AB_466805 | FC (1:300) |
| Antibody | Biotin anti-mouse CD3ε (Armenian Hamster monoclonal) | eBioscience | Cat# 13-0031-82 RRID:AB_466319 | FC (1:300) |
| Antibody | PE anti-rat CD2 (Mouse monoclonal) | BioLegend | Cat# 201305 RRID:AB_2073811 | FC (1:500) |
| Antibody | PE anti-human NGFR (Mouse monoclonal) | eBioscience | Cat# 12-9400-42 RRID:AB_2572710 | FC (1:500) |
| Antibody | PE Hamster IgG (Armenian Hamster monoclonal) | BioLegend | Cat# 400907 RRID:AB_326593 | FC (1:200) |
| Antibody | PE anti-mouse Notch1 (Armenian Hamster monoclonal) | BioLegend | Cat# 130607 RRID:AB_1227719 | FC (1:200) |
| Antibody | PE anti-mouse Notch2 (Armenian Hamster monoclonal) | BioLegend | Cat# 130707 RRID:AB_1227725 | FC (1:200) |
| Antibody | PE anti-mouse Notch3 (Armenian Hamster monoclonal) | BioLegend | Cat# 130507 RRID:AB_1227733 | FC (1:200) |
| Antibody | PE anti-mouse Notch4 (Armenian Hamster monoclonal) | BioLegend | Cat# 128407 RRID:AB_1133997 | FC (1:200) |

*Continued on next page*

*Continued*

| Reagent type (species) or resource | Designation | Source or reference | Identifiers | Additional information |
|---|---|---|---|---|
| Antibody | Alexa Fluor 647 Mouse IgG1 (Mouse monoclonal) | BioLegend | Cat# 400130 RRID:AB_2800436 | FC (2 µl per test) |
| Antibody | Alexa Fluor 647 anti-GATA3 (Mouse monoclonal) | BD Biosciences | Cat# 560068 RRID:AB_1645316 | FC (8 µl per test) |
| Antibody | Rabbit-IgG (Rabbit monoclonal) | Cell Signaling Technology | Cat# 3900 RRID:AB_1550038 | FC (1:500) |
| Antibody | Anti-TCF1 (Rabbit monoclonal) | Cell Signaling Technology | Cat# 2203 RRID:AB_2199302 | FC (1:100) |
| Antibody | Anti-LMO2 (Rabbit monoclonal) | Abcam | Cat# ab91652 RRID:AB_2049879 | FC (1:200) ChIP (2ug/sample) |
| Antibody | Anti-c-Myc (Rabbit monoclonal) | Cell Signaling Technology | Cat# 5605 RRID:AB_1903938 | FC (1:100) |
| Antibody | DyLight 488 anti-rabbit IgG (Donkey polyclonal) | BioLegend | Cat# 406404 RRID:AB_1575130 | FC (1:250) |
| Antibody | DyLight 649 anti-rabbit IgG (Donkey polyclonal) | BioLegend | Cat# 406406 RRID:AB_1575135 | FC (1:500) |
| Antibody | Anti-Tubulinα (Mouse monoclonal) | Sigma | Cat# T6199 RRID:AB_477583 | WB (1:1000) |
| Antibody | Anti-LMO2 (Mouse monoclonal) | Novus | Cat# NB110-78626 RRID:AB_1084895 | WB (1:1000) ChIP (2 µg/sample) |
| Antibody | Anti-human LMO2 (Goat polyclonal) | R&D Systems | Cat# AF2726 RRID:AB_2249968 | ChIP (2 µg/sample) |
| Antibody | Anti-H3K4me3 (Rabbit polyclonal) | Millipore | Cat# 07-473 RRID:AB_1977252 | ChIP (5 µl/sample) |
| Recombinant DNA reagent (plasmid) | pCMV-VSV-G-RSV-Rev | RIKEN BRC | Cat# RDB04393 | Lentiviral packaging plasmid |
| Recombinant DNA reagent (plasmid) | pCAG-HIVgp | RIKEN BRC | Cat# RDB04394 | Lentiviral packaging plasmid |
| Recombinant DNA reagent (plasmid) | pLVS-EF-IR2 | This paper | | Lentiviral vector with IRES-rat CD2 |
| Recombinant DNA reagent (plasmid) | mLmo2/pLVS-EF-IR2 | This paper | | pLVS-EF-IR2 Lentiviral vector encoding mLmo2 |
| Recombinant DNA reagent (plasmid) | mTcf7/pLVS-EF-IR2 | This paper | | pLVS-EF-IR2 Lentiviral vector encoding mTcf7 |
| Recombinant DNA reagent (plasmid) | GCDN | *Hirano et al., 2015* | | Retroviral vector with IRES-human NGFR |
| Recombinant DNA reagent (plasmid) | hBCL2/GCDN | This paper | | GCDN Retroviral vector encoding hBCL2 |
| Recombinant DNA reagent (plasmid) | mMeis1/GCDN | This paper | | GCDN Retroviral vector encoding mMeis1 |

*Continued on next page*

*Continued*

| Reagent type (species) or resource | Designation | Source or reference | Identifiers | Additional information |
|---|---|---|---|---|
| Recombinant DNA reagent (plasmid) | mHmga2/GCDN | This paper | | GCDN Retroviral vector encoding mHmga2 |
| Recombinant DNA reagent (plasmid) | Cas9-GFP | *Hosokawa et al., 2018* | | Retroviral vector to express Cas9 and GFP |
| Recombinant DNA reagent (plasmid) | E42-dTet | *Hosokawa et al., 2018* | | Retroviral vector to express sgRNA and human NGFR |
| Recombinant DNA reagent (plasmid) | sgRNA against Luciferase (control) | *Hosokawa et al., 2018* | | 5'-ggcatttcgcag cctaccg-3' |
| Recombinant DNA reagent (plasmid) | sgRNA against LMO2 #1 | This paper | | 5'-tcgatggccgag gacattg-3' |
| Recombinant DNA reagent (plasmid) | sgRNA against LMO2 #2 | This paper | | 5'-aatgtcctcggc catcgaa-3' |
| Recombinant DNA reagent (plasmid) | sgRNA against LMO2 #3 | This paper | | 5'-gaaagccatcga ccagtac-3' |
| Sequence-based reagent | ActB (Forward) | This paper | PCR primers | 5'-tacagcccgggg agcat-3' |
| Sequence-based reagent | ActB (Reverse) | This paper | PCR primers | 5'-acacccgccac cagttc-3' |
| Sequence-based reagent | Meis1 (Forward) | This paper | PCR primers | 5'-gacgctttaaag agagataaagatgc-3' |
| Sequence-based reagent | Meis1 (Reverse) | This paper | PCR primers | 5'- catttctcaaa aatcagtgctaaga -3' |
| Sequence-based reagent | Hmga2 (Forward) | This paper | PCR primers | 5'-aaggcagcaaaa acaagagc-3' |
| Sequence-based reagent | Hmga2 (Reverse) | This paper | PCR primers | 5'-gccgtttttctc caatggt-3' |
| Sequence-based reagent | Bcl11a (Forward) | This paper | PCR primers | 5'-ccaaacaggaac acacatagcaga-3' |
| Sequence-based reagent | Bcl11a (Reverse) | This paper | PCR primers | 5'-ggggattagagc tccgtgt-3' |
| Sequence-based reagent | Gata3 (Forward) | This paper | PCR primers | 5'-ttatcaagccca agcgaag-3' |
| Sequence-based reagent | Gata3 (Reverse) | This paper | PCR primers | 5'-tggtggtggtct gacagttc-3' |
| Sequence-based reagent | Lmo2 (Forward) | This paper | PCR primers | 5'-gaggcgcctcta ctacaa-3' |
| Sequence-based reagent | Lmo2 (Reverse) | This paper | PCR primers | 5'-gatccgcttgt cacaggatg-3' |
| Sequence-based reagent | Tcf7 (Forward) | This paper | PCR primers | 5'-cagctcccccc atactgtgag-3' |
| Sequence-based reagent | Tcf7 (Reverse) | This paper | PCR primers | 5'-tgctgtctatat ccgcaggaa-3' |
| Sequence-based reagent | Tcf7 promoter region (Forward) | This paper | PCR primers | 5'-ttaagtttta ttggtgaatgagtt-3' |
| Sequence-based reagent | Tcf7 promoter region (Reverse) | This paper | PCR primers | 5'-aaaaaactccaa aaataaaacccac-3' |

*Continued on next page*

*Continued*

| Reagent type (species) or resource | Designation | Source or reference | Identifiers | Additional information |
|---|---|---|---|---|
| Sequence-based reagent | Tcf7 TSS (Forward) | This paper | PCR primers | 5'-gcagcaagg gttgcattt-3' |
| Sequence-based reagent | Tcf7 TSS (Reverse) | This paper | PCR primers | 5'-ttgtctgtactg ggctgtttacat-3' |
| Sequence-based reagent | Tcf7 -31kb (Forward) | This paper | PCR primers | 5'-ttccatccac cgttttgttt-3' |
| Sequence-based reagent | Tcf7 -31kb (Reverse) | This paper | PCR primers | 5'-ggcgtgtggt gggaatacta-3' |
| Sequence-based reagent | Tcf7 -35kb (Forward) | This paper | PCR primers | 5'-ctgcaagc agctggaagtc-3' |
| Sequence-based reagent | Tcf7 -35kb (Reverse) | This paper | PCR primers | 5'-cactggaagctg tgagtgatg-3' |
| Sequence-based reagent | Igk 3'UTR (Forward) | This paper | PCR primers | 5'-ggcacatc tgttgctttcgc -3' |
| Sequence-based reagent | Igk 3'UTR (Reverse) | This paper | PCR primers | 5'-ggggtaggga gcaggtgtat-3' |
| Peptide, recombinant protein | PerCpCy5.5 streptavidin | BioLegend | Cat# 405214 | FC (1:200) |
| Peptide, recombinant protein | Recombinant Mouse SCF | PeproTech | Cat# 250-03 | |
| Peptide, recombinant protein | Recombinant Human FLT3L | PeproTech | Cat# 300-19 | |
| Peptide, recombinant protein | Recombinant Mouse IL-7 | PeproTech | Cat# 217-17 | |
| Commercial assay or kit | Foxp3 / Transcription Factor Staining Set | eBioscience | Cat# 00-5523-00 | Used to detect TCF1 and GATA3 |
| Commercial assay or kit | Fixation/ Permeabilization Solution Kit with BD GolgiStop | BD Biosciences | Cat# 554715 | Used to detect Lmo2 and c-Myc |
| Commercial assay or kit | Permeabilization Buffer Plus | BD Biosciences | Cat# 561651 | Used to detect Lmo2 and c-Myc |
| Commercial assay or kit | High Capacity cDNA Reverse Transcription Kit | Thermo Fisher Scientific | Cat# 4368814 | |
| Commercial assay or kit | NucleoSpin Tissue | TaKaRa Bio | Cat# 740952.50 | |
| Commercial assay or kit | MethylEasy Xceed | TaKaRa Bio | Cat# ME002 | |
| Commercial assay or kit | TaKaRa EpiTaq HS (for bisulfite-treated DNA) | TaKaRa Bio | Cat# R110A | |
| Commercial assay or kit | Mighty TA-cloning kit | TaKaRa Bio | Cat# 6028 | |
| Commercial assay or kit | Anti-Biotin MicroBeads | Miltenyi Biotec | Cat# 130-090-485 | |
| Commercial assay or kit | Dynabeads Protein A | Thermo Fisher Scientific | Cat# 10001D | |

*Continued on next page*

*Continued*

| Reagent type (species) or resource | Designation | Source or reference | Identifiers | Additional information |
|---|---|---|---|---|
| Commercial assay or kit | Dynabeads Protein G | Thermo Fisher Scientific | Cat# 10003D | |
| Commercial assay or kit | Dynabeads M-280 Sheep Anti-Rabbit IgG | Thermo Fisher Scientific | Cat# 11203D | |
| Commercial assay or kit | NE-PER Nuclear and Cytoplasmic Extraction Reagents | Pierce | Cat# 78833 | |
| Commercial assay or kit | PCR purification Kit | Qiagen | Cat# 28004 | |
| Commercial assay or kit | Fast SYBR Green Master Mix | Thermo Fisher Scientific | Cat# 4385614 | |
| Commercial assay or kit | NEBNext Ultra II DNA Library Prep with Sample Purification Beads | NEB | Cat# E7103S | |
| Commercial assay or kit | NEBNext Multiplex Oligos for Illumina | NEB | Cat# E7500S | |
| Chemical compound, drug | Trizol reagent | Thermo Fisher Scientific | Cat# 15596026 | |
| Chemical compound, drug | 7-AAD | BioLegend | Cat# 420403 | FC (1:50) |
| Chemical compound, drug | Alexa Fluor 647 Annexin V | BioLegend | Cat# 640911 | FC (1:40) |
| Chemical compound, drug | DSG (disuccinimidyl glutarate) | Thermo Fisher Scientific | Cat# 20593 | 1 mg/ml |
| Software, algorithm | FlowJo | BD Biosciences | RRID:SCR_008520 | |

## Mice

*Ebf1$^{+/}$* mice were provided by R. Grosschedl (Max Planck Institute of Immunobiology and Epigenetics). *Ebf1*-deficient embryos were generated from *Ebf1$^{+/}$* intercrosses. https://www.jax.org/strain/002320 B6.Cg-Tg(BCL2)25Wehi/J (Bcl2-Tg), B6.Gt(ROSA)26$^{Sortm1.1(CAG-cas9*,-EGFP)Fezh}$/J (Cas9 knock-in), B6.Cg-Rag2$^{tm1.1Cgn}$/J (RAG2 KO), and B6.129S4-Il2rg$^{tm1Wjl}$/J (Cγ KO) mice were purchased from the Jackson Laboratory. All mice were maintained in specific pathogen free conditions, and animal experiments were approved by the Animal Experimentation Committee (Tokai University, Kanagawa, Japan).

## Flow cytometry

For flow cytometric analysis, the following monoclonal antibodies (mAbs) and reagents were used: CD8a (53–6.7), CD11b (M1/70), CD44 (IM7), CD45 (30-F11), CD45.1 (A20), CD45.2 (104), B220 (RA3-6B2), Thy1.2 (30-H12), Gr-1 (RB6-8C5), rat-CD2 (OX34), Hamster IgG (HTK888), Notch1 (HMN1-12), Notch2 (HMN2-35), Notch3 (HMN3-133), Notch4 (HMN4-14), anti-Rabbit-IgG (Poly4064), mIgG1 (MOPC-21), AnnexinV and 7-AAD were purchased from BioLegend. CD25 (PC61.5), CD45 (30-F11), TER119 (TER-119), Gr-1 (RB6-8C5), CD11b (M1/70), CD11c (N418), CD19 (1D3), NK1.1 (PK136), CD3e (145–2 C11), hNGFR (ME20.4) were purchased from eBioscience. CD4 (GK1.5, RM4-5) and GATA3 (L50-823) were purchased from BD Biosciences. CD19 (1D3) was purchased from Tonbo Biosciences. TCF1 (C63D9), c-Myc (D84C12) and Rabbit-IgG (DA1E) were purchased from Cell Signaling Technology. LMO2 (EP3257) was purchased from Abcam. For intracellular staining, cells were fixed and permeabilized with Foxp3/Transcription Factor Staining

Set (eBioscience) or Fixation/Permeabilization Solution Kit with BD GolgiStop (BD Biosciences) and Permeabilization Buffer Plus (BD Biosciences). Stained cells were measured with FACSCalibur (BD Biosciences) or FACSVerse (BD Biosciences). Data were analyzed using FlowJo (BD Biosciences).

## Isolation of hematopoietic progenitors and establishment of pro-B(+) and pro-B(–) cell lines

For the establishment of HPC lines, *Ebf1*-deficient progenitor cells were isolated from *Ebf1*-deficient E14.5 fetal liver (both male and female). FL cells were incubated with biotin-conjugated mAbs against lineage markers (TER119 and Gr-1, Lin). The cells were then washed and incubated with anti-biotin Microbeads (Miltenyi Biotec), and Lin cells were enriched using autoMACS (Miltenyi Biotec). Lin⁻ cells were cultured on OP9 or TD7 cells in the presence of mSCF, hFlt3L and mIL7. We established two pro-B cell lines, one that was maintained on OP9 cells was called as pro-B(+) cells, and the other that was maintained on TD7 cells was called as pro-B(–) cells.

## Cell lines and cultures

TD7, thymic stromal cell line established from fetal thymus (B6, embryonic day 15.5) was cultured in RPMI 1640 (Nissui Pharmaceutical Co., Tokyo)with 10% FBS, sodium pyruvate (Sigma), L-glutamine (Wako), penicillin (Meiji Seika Pharma), streptomycin (Meiji Seika Pharma), 2-ME (Gibco). These cell line seems to be derived from thymic mesenchymal cells as they express PDGFRα. OP9 cell line was kindly provided from Dr. K. Yokoyama (University of Tokyo, *Yokoyama et al., 2013*) and cultured in α-MEM (Wako) with 20% FBS, penicillin, streptomycin, 2-ME. This cell line kept features in common with OP9-K (RRID:CVCL_KB57). These cell lines were confirmed to be mycoplasma-free status before experiments. Pro-B cell lines were cultured in IMDM (Wako) with 10% FBS, penicillin, streptomycin, 2-ME, 10 ng/ml mSCF (Peprotech), 10 ng/ml hFlt3L (Peprotech), 10 ng/ml mIL7 (Peprotech) on TD7 or mitomycin C (Kyowa-Kirin)-treated OP9. For T-cell induction, pro-B cells were co-cultured on OP9-Dll4 for 3–7 days under the same conditions as the maintenance of pro-B cell lines.

## Viral vector transduction of pro-B cell lines

Retroviral vector GCDN (mock vector containing IRES-hNGFR) (*Hirano et al., 2015*), encoding *hBCL2*, *mMeis1*, and *mHmga2* were generated by transient transfection into PLAT-E packaging cells. Lentiviral vector pLVS-EF-IR2 (mock vector containing IRES-rat CD2), encoding *mLmo2* and *mTcf7* were generated by transient transfection into 293T as described previously (*Hirano et al., 2020*). Pro-B cells were transduced with viral supernatants by spin infection as described previously (*Hozumi et al., 2003*). After the infection, cells were plated and maintained on stromal cells.

## RNA preparation and RT-qPCR

Total RNA was isolated from $4510^5$ cells using Trizol reagent (Thermo Fisher Scientific) according to the manufacturer's instructions. cDNA was synthesized with High Capacity cDNA Reverse Transcription Kit (Thermo Fisher Scientific). Quantitative PCR (qPCR) was performed with Fast SYBR Green Master Mix (Thermo Fisher Scientific) and LightCycler 480 System II (Roche). The following primer sets were used for qPCR are listed below.

> *ActB*, 5'-tacagcccgggggagcat-3' and 5'-acacccgccaccagttc-3'
> *Meis1*, 5'-gacgctttaaagagagagataaagatgc-3' and 5'- catttctcaaaaatcagtgctaaga −3'
> *Hmga2*, 5'-aaggcagcaaaaacaagagc-3' and 5'-gccgtttttctccaatggt-3'
> *Bcl11a*, 5'-ccaaacaggaacacacatagcaga-3' and 5'-ggggattagagctccgtgt-3'
> *Gata3*, 5'-ttatcaagcccaagcgaag-3' and 5'-tggtggtggtctgacagttc-3'
> *Lmo2*, 5'-gaggcgcctctactacaa-3' and 5'-gatccgcttgtcacaggatg-3'
> *Tcf7*, 5'-cagctcccccatactgtgag-3' and 5'-tgctgtctatatccgcaggaa-3'.

## Microarray analysis

Total RNA was extracted from *Ebf1*-deficient pro-B(–) and pro-B(+) cells, and each 100 ng RNA was used for microarray analysis. Microarray experiments were performed with Whole Mouse Genome 4x44K ver. 2 Microarray Kit (Agilent) using the Agilent One-Color Microarray-Based Gene Expression Analysis protocol. Data from samples that passed the QC parameters were subjected to 75th percentile normalization and analyzed using Genespring GX (version 12, Agilent Technologies).

CRISPR/Cas9-mediated deletion of LMO2 in Pro-B cell lines sgRNA expression vector (E42-dTet) and Cas9-GFP expression vector were described previously (*Hosokawa et al., 2018*). 19-mer sgRNAs were designed using the Benchling web tool (https://www.benchling.com) and inserted into the empty sgRNA-expression vector by PCR-based insertion. Three sgRNA-expression vectors were generated for one gene, and pooled retroviral plasmids were used to make retroviral supernatant. Sequences of sgRNAs used in this study are listed below.

Control (Luciferase); ggcatttcgcagcctaccg
LMO2 #1; tcgatggccgaggacattg
LMO2 #2; aatgtcctcggccatcgaa
LMO2 #3; gaaagccatcgaccagtac

Pro-B cell lines were transduced with retroviral vectors encoding Cas9-GFP, and 3 days after infection, GFP$^+$ retrovirus-infected cells were sorted. Then, they were expanded for a week and subjected second retrovirus transduction with sgLMO2-hNGFR. They were transferred onto OP9-Dll4 on day5 or day10 after snd infection, then CD25 and CD44 profiles on GFP$^+$ hNGFR$^+$ retrovirus-infected cells were analyzed, 3 days later.

## CRISPR/Cas9-mediated deletion of LMO2 in BM progenitors

BM was obtained from the femurs and tibiae of 2–3 monthold Cas9 and Bcl2 Tg mice. Suspensions of BM cells were prepared and stained for lineage (Lin) markers using biotin-conjugated lineage antibodies CD11b (eBioscience; 13-0112-86), CD11c (eBioscience; 13-0114-85), Gr-1 (eBioscience; 13-5931-86), TER-119 (eBioscience; 13-5921-85), NK1.1 (eBioscience; 13-5941-85), CD19 (eBioscience; 13-0193-85), and CD3ε (eBioscience; 13-0031-082); incubated with streptavidin-coated magnetic beads (Miltenyi Biotec); and passed through a magnetic column using AutoMACS with the 'Depelete' program (Miltenyi Biotec). Thereafter, the hematopoietic progenitors were transduced with retroviral vectors encoding sgRNA against LMO2 and cultured using the OP9 medium (α-MEM, 20% FBS, 50 μM β-mercaptoethanol, Pen-Strep-Glutamine) supplemented with 10 ng/ml of IL-7 (Pepro Tech Inc) and 10 ng/ml of SCF (Pepro Tech Inc). Two days after the sgLMO2 transduction, the cells were collected and cultured on OP9-Dll1 monolayers using OP9 medium supplemented with 10 ng/ml of IL-7 and 10 ng/ml of Flt3L (Pepro Tech Inc) for 4 days. The cultured cells were then disaggregated, filtered through a 40 μm nylon mesh, and subjected to flow cytometry analysis using surface antibodies against CD45 PECy7 (eBioscience; 25-0451-82), CD44 FITC (BioLegend; 103005), CD25 APC-e780 (eBioscience; 47-0251-82), human-NGFR PE (eBioscience; 12-9400-42), and a biotin-conjugated lineage cocktail (CD8α, CD11b, CD11c, Gr-1, TER-119, NK1.1, CD19, TCRβ, and TCRγδ) with streptavidin PerCPCy5.5.

## Immunoblotting

Cytoplasmic and nuclear extracts, used to the detection of Tubulinα and LMO2, respectively, were prepared by using NE-PER Nuclear and Cytoplasmic Extraction Reagents (Pierce). Lysates were run on 12.5% polyacrylamide gel, followed by immunoblotting. The antibodies used for the immunoblot analysis were anti-Tubulinα (Sigma, T6199) and anti-LMO2 (Novus, NB110-78626).

## DNA methylation analysis

Genomic DNA was isolated from pro-B(+), pro-B(–) and LMO2/pro-B(−) cells. The DNA was purified using NucleoSpin Tissue (TaKaRa Bio) and was treated with bisulfite using MethylEasy Xceed (TaKaRa Bio) according to the manufacturer's instructions. For the evaluation of DNA methylation status of CpG island in *Tcf7* promoter region, which contains 25 potential CpG methylation sites, bisulfite-converted DNA was amplified by PCR using TaKaRa EpiTaq HS (for bisulfite-treated DNA, TaKaRa Bio) and the following primer set: 5'-ttaagtttttattggtgaatgagtt-3' and 5'-aaaaaactccaaaaataaaacccac-3'. Amplified PCR products were cloned into pMD20-T vector using Mighty TA-cloning kit (TaKaRa Bio) and then sequenced.

## Chromatin immunoprecipitation (ChIP) and ChIP-sequencing

$1 \times 10^7$ cells were fixed with 1 mg/ml DSG (Thermo Scientific) in PBS for 30 min at RT followed by an additional 10 min with addition of formaldehyde up to 1%. The reaction was quenched by addition of 1/10 vol of 0.125 M glycine and the cells were washed with HBSS (Gibco). Pelleted nuclei

were dissolved in lysis buffer (0.5% SDS, 10 mM EDTA, 0.5 mM EGTA, 50 mM TrisHCl [pH 8] and PIC) and sonicated on SFX150 (Branson) for scycles of 20 s sonication followed by 1 min rest, with 30% amplitude. Six mof anti-LMO2 Abs (a mixture of 2 µg of NB110-78626 [Novus], 2 µg of ab91652 [Abcam] and 2 µg of AF2726 [R and D systems]) or anti-H3K4-3Me (07–043, Millipore) were pre-bound to Dynabeads Protein A/G or Dynabeads anti-Rabbit Ig (Invitrogen) and then added to the diluted chromatin complexes. The samples were incubated overnight at 4℃, then washed and eluted for 6 hr at 65℃ in ChIP elution buffer (20 mM Tris–HCl, pH 7.5, 5 mM EDTA, 50 mM NaCl, 1% SDS, and 50 µg/ml proteinase K). Precipitated chromatin fragments were cleaned up using PCR purification Kit (Qiagen).

Quantitative PCR analysis was performed on QuantStudio3 (Applied Biosystems) using the Fast SYBR Green Master Mix. Data are shown as mean values (% input). The primers used are listed below:

> *Tcf7* TSS, 5'-gcagcaagggttgcattt-3' and 5'-ttgtctgtactgggctgtttacat-3'
> *Tcf7*-31kb, 5'-ttccatccaccgttttgttt-3' and 5'-ggcgtgtggtgggaatacta-3'
> *Tcf7*-35kb, 5'-ctgcaagcagctggaagtc-3' and 5'-cactggaagctgtgagtgatg-3'
> *Igk* 3' UTR, 5'-ggcacatctgttgctttcgc −3' and 5'-ggggtagggagcaggtgtat-3'

ChIP-seq libraries were constructed using NEBNext Ultra II DNA Library Prep with Sample Purification Beads (E7103S, NEB) and NEBNext Multiplex Oligos for Illumina (E7500S, NEB) and sequenced on Illumina NextSeq500 in single read mode with the read length of 75 nt. Base calls were performed with RTA 1.13.48.0 followed by conversion to FASTQ with bcl2fastq 1.8.4 and produced approximately 30 million reads per sample. ChIP-seq data were mapped to the mouse genome build NCBI37/mm10 using Bowtie (v1.1.1; http://bowtie-bio.sourceforge.net/index.shtml) with '-v 3 k 11 m 10 t `-best` –strata' settings and HOMER tagdirectories were created with *make-TagDirectory* and visualized in the UCSC genome browser (http://genome.ucsc.edu). ChIP peaks were identified with *findPeaks.pl* against a matched control sample using the settings '-P. 1 -LP. 1 -poisson. 1 -style factor'. The identified peaks were annotated to genes with the *annotatePeaks.pl* command against the mm10 genomic build in the HOMER package. Peak reproducibility was determined by a HOMER adaptation of the IDR (Irreproducibility Discovery Rate) package according to ENCODE guidelines (https://sites.google.com/site/anshulkundaje/projects/idr). Only reproducible high-quality peaks, with a normalized peak score ≥ 15, were considered for further analysis. Motif enrichment analysis was performed with the *findMotifsGenome.pl* command in the HOMER package using a 200 bp window.

## Acknowledgements

We thank Drs. R Goitsuka and K Nakashima for providing the expression vector for murine *Meis1* and *Hmga2*, respectively; Dr. R Grosshedl for giving *Ebf1*[+/] mice; Dr. K Yokoyama for providing OP9 cell line and critical advice for its culture conditions; Drs. J Takano and T Ikawa for technical advice and support; N Abe S Ochiai for experimental assistance and members of the Support Center for Medical Research and Education at Tokai University for their technical help. This work was partly supported by the Tokai University General Research Organization Research and Study Project. We would like to thank Editage for English editing.

## Additional information

### Funding

| Funder | Grant reference number | Author |
| --- | --- | --- |
| Japan Society for the Promotion of Science | 16K08848 | Katsuto Hozumi |
| Japan Society for the Promotion of Science | JP19H03692 | Hiroyuki Hosokawa |
| Japan Society for the Promotion of Science | 17H05802 | Katsuto Hozumi |
| Japan Society for the Promo- | JP20K07730 | Katsuto Hozumi |

tion of Science

| Uehara Memorial Foundation | Research grant | Hiroyuki Hosokawa |
|---|---|---|
| Naito Foundation | Research grant | Hiroyuki Hosokawa |
| Takeda Science Foundation | Research grant | Hiroyuki Hosokawa |
| Yasuda Medical Foundation | Research grant | Hiroyuki Hosokawa |
| SENSHIN Medical Research Foundation | Research grant | Hiroyuki Hosokawa |
| Daiichi Sankyo Foundation of Life Science | Research grant | Hiroyuki Hosokawa |
| Tokyo Biochemical Research Foundation | Research grant | Hiroyuki Hosokawa |
| Princess Takamatsu Cancer Research Fund | Research grant | Hiroyuki Hosokawa |
| Mitsubishi Foundation | Research grant | Hiroyuki Hosokawa |
| Tokai University School of Medicine Research Aid | Research grant | Ken-ichi Hirano Hiroyuki Hosokawa Maria Koizumi |

The funders had no role in study design, data collection and interpretation, or the decision to submit the work for publication.

## Author contributions

Ken-ichi Hirano, Conceptualization, Formal analysis, Investigation, Visualization, Writing - original draft, Writing - review and editing; Hiroyuki Hosokawa, Conceptualization, Data curation, Formal analysis, Funding acquisition, Validation, Investigation, Visualization, Writing - original draft, Writing - review and editing; Maria Koizumi, Formal analysis, Investigation; Yusuke Endo, Takashi Yahata, Formal analysis, Validation, Investigation, Methodology; Kiyoshi Ando, Supervision, Validation; Katsuto Hozumi, Conceptualization, Data curation, Formal analysis, Supervision, Funding acquisition, Validation, Investigation, Visualization, Methodology, Writing - original draft, Project administration, Writing - review and editing

## Author ORCIDs

Ken-ichi Hirano https://orcid.org/0000-0003-3495-610X
Hiroyuki Hosokawa https://orcid.org/0000-0002-9592-2889
Maria Koizumi http://orcid.org/0000-0001-8275-5594
Katsuto Hozumi https://orcid.org/0000-0002-7685-6927

## Ethics

Animal experimentation: This study was performed in strict accordance with the recommendations in the Guidelines for the Care and Use of Animals for Scientific Purposes at Tokai University, and approved by the Animal Experimentation Committee of Tokai University (Approval No.: 165015, 171002, 182026, 193040, 204028, 211006), which is further monitored by the Animal Experimentation Evaluation Committee of Tokai University with researcher for Humanities/Sociology and external expert.

## Decision letter and Author response

Decision letter https://doi.org/10.7554/eLife.68227.sa1
Author response https://doi.org/10.7554/eLife.68227.sa2

# Additional files

## Supplementary files

• Supplementary file 1. Microarray analysis of pro-B(+) and pro-B(−) cells.

• Transparent reporting form

### Data availability

Microarray expression data from Ebf1-deficient pro-B(-) and pro-B(+) cells, and the new deep-sequencing data reported in this paper are available via GEO (https://www.ncbi.nlm.nih.gov/geo/) (GSE162549 and GSE154472).

The following datasets were generated:

| Author(s) | Year | Dataset title | Dataset URL | Database and Identifier |
|---|---|---|---|---|
| Hozumi K, Hirano Ki | 2020 | Difference in gene expression signatures of Ebf1-deficient pro-B cell lines with and without the ability to differentiate into T cell lineage | https://www.ncbi.nlm.nih.gov/geo/query/acc.cgi?acc=GSE162549 | NCBI Gene Expression Omnibus, GSE162549 |
| Hirano Ki, Hosokawa H, Koizumi M, Endo Y, Hozumi K | 2020 | LMO2 is essential to maintain the ability of progenitors to differentiate into T-cell lineage in mice | https://www.ncbi.nlm.nih.gov/geo/query/acc.cgi?acc=GSE154472 | NCBI Gene Expression Omnibus, GSE154472 |

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
