## [Decision Letter]

**Acceptance summary:**

The work presented in this paper investigates the function of a transcription factor, LMO2, during T cell development. The results provided likely have an important impact for our understanding of the mechanisms driving the development of some leukemia, and may also apply to normal hematopoiesis.

**Decision letter after peer review:**

Thank you for submitting your article "Lmo2 is essential to maintain the ability of progenitors to differentiate into T-cell lineage" for consideration by *eLife*. Your article has been reviewed by 3 peer reviewers, including JC Zúñiga-Pflücker as the Reviewing Editor and Reviewer #1, and the evaluation has been overseen by Satyajit Rath as the Senior Editor. The following individuals involved in review of your submission have agreed to reveal their identity: Christelle Harly (Reviewer #2); Trang Hoang (Reviewer #3).

Essential revisions:

1) A better characterization of how the proB cells were derived.

2) A characterization of the differentiation potential of the proB cells cultured on OP9-DL4 that shows their ability to go beyond the DN2 stage of T cell development.

3) Demonstration of the timing of histone modifications after CRISPR/Cas9 induced loss of Lmo2 expression, which would help explain the observed delay in the loss of T cell potential.

4) Cell death calculations need to be redone as % cell death.

*Reviewer #1:*

The work by Hirano et al. addresses an important and not fully resolved issue regarding the regulation of the initiation of T cell lineage commitment. The approach used employs an Ebf1-deficient pro-B cell line system that can be induced to differentiate toward the T cell lineage after receiving strong Notch signaling in vitro. However, a subset of these pro-B cells failed to give rise to T-lineage cells, which was attributed to low levels of Lmo2 expression as compared to the pro-B cells that retain T-lineage differentiation potential. Enforced expression of Lmo2 rescued the defect, which was shown to act by regulating epigenetic changes at the Tcf7 gene locus. However, what remains unclear is whether Lmo2 is required at the start of T-cell lineage differentiation to enable entry into the program, as well as serving to facilitate the elaboration of the program as the cells commit to the T cell lineage. Nevertheless, even without clarity as to the cellular subset or subsets requiring Lmo2 function, the present finds provide a clear rationale for considering Lmo2 as an important player during early T cell differentiation.

A few concerns, when addressed, would improve the impact of the findings.

1. It is clear that the proB(-) do not fare well in the Dll4+ cell cultures, and that the proB(+) cells can differentiate and survive well in the Dll4+ cell cultures, however, with only CD44 and CD25 expression as the only indicators of differentiation, and CD44 is expressed on both culture conditions, it would be good to also show an additional T-lineage related marker, e.g., CD90 to further support their differentiation as T cells. Additionally, do the proB(+) cells differentiate to later stages of T cell development, ie., do they eventually express CD3, CD4, CD8 and TCRs?

2. The results shown in Figure 2 make it seem that in the acute absence of Lmo2 the effect is not in the initiation of T-lineage commitment and early differentiation but rather affecting their progression or transition to later stages. Perhaps the authors should consider this interpretation of their results in the discussion and/or Results section.

3. The Supplemental Excel file with the microarray results appears to be missed labeled, i.e., reversed heading for ProB(-) vs ProB(+), as the data shown are the opposite to what is summarized in Figure 1.

4. The survival/death analysis shown in Figure 3, supplement 1, appears to show an increase in cell death in proB(-) cells when cultured on Dll4+ cells, while the ectopic expression of Bcl2 would appear to lessen the number of dying cells. However, if one were to use the ratio of dead cells from Dll4 vs Mock cell cultures (Dll4/Mock) to determine the fold increase in the % of cells dying in each condition, then the number would be as follows: 2x for proB(+); 6.4x for proB(-)/Mock; and 7.3x for proB(-)/Bcl2, which would lead to the conclusion that Bcl2 is not rescuing the cells from dying, and I feel that this is the more likely and correct interpretation of the results. Minimally, the authors should convey this notion in the results.

5. In Figure 4c, the results are quite striking and beautifully demonstrate that ectopic expression TCF1 in proB(-)/Bcl2 cells enables their ability to differentiate into CD25+ cells. However, it is not clear whether this would require Dll4-dependent signals as the Mock cell cultures are not shown. The authors should include this experiment, as it provides insights as to the requirement for Notch signaling in cells that have enforced TCF1 expression, which have been shown by the Bhandoola lab and others.

6. Lastly, some of the discussion statements, in page 18, which argue for a role for Lmo2 in prethymic progenitors are not fully backed by the results presented in the paper, as prethymic progenitors were not examined, these conclusions should be tempered as possible rather as having been shown.

*Reviewer #2:*

In this manuscript, Hirano et al. investigate the mechanisms that maintain T cell potential in pre-thymic lymphoid precursors. The authors establish a new stromal cell lines from fetal thymus, called TD7. They use this cell line to establish an Ebf1-deficient pro-B cell line (called pro-B(-)), using a protocol similar to the one previously described to generate Ebf1-deficient pro-B cells using the well-established OP9 stromal cell line (called pro-B(+)). Surprisingly, they find that unlike pro-B(+), pro-B(-) fail to engage toward the T cells lineage in response to Notch signaling. To understand this puzzling difference, they transcriptionally profile the two pro-B cell lines, and find that the transcription factor Lmo2 is expressed at lower level on pro-B(-) compared to pro-B(+). They propose that Lmo2 plays important functions to enable expression of the transcription factor TCF-1 in response to Notch signaling (encoded by Tcf7). To support this model, they notably show that Lmo2 overexpression rescues the T cell potential of pro-B(-), and conversely, Lmo2 loss of function restrain the T cell potential of pro-B(+). Importantly, they show that TCF-1 ectopic expression rescue the T cell potential of pro-B(-). They propose that Lmo2 may play key functions in maintaining T cell potential in pre-thymic T cell precursors.

In this elegant study, the authors provide a detailed characterization of the effect of ectopically expressing or deleting Lmo2 on transcription and epigenetics in Ebf1-deficient B cell lines, and their differentiation toward the T cell lineage. The results provided likely have an important impact for our understanding of the mechanisms driving the development of Lmo2-induced T cell leukemia. However, it is unclear whether this model also apply to normal hematopoiesis.

I believe that this very interesting study would be greatly strengthened and its impact increased if the authors could address the following points:

1) The novel TD7 stromal cell line and the pro-B cells generated on this cell line are poorly characterized. The authors state that "We established our original thymic stromal cell line, TD7, in which HSCs differentiate into CD19^+^ B-lineage cells in the presence of IL-7 or into Thy1+CD25+ T lineage cells upon receiving Notch signaling.", "the establishment of pro-B cells on TD7 was accompanied by a prolonged emergence of myeloid marker-positive cells compared to that on OP9, and the remaining pro-B cells lost their pluripotency". However, this data is shown. The careful characterization of the cells generated on TD7 would be very valuable for the readers, and key to properly interpret the data presented in the manuscript. It is presently unclear how the pro-B(-) are related to pro-B cells. The authors mention that pro-B(-) cells express B220 but this is not shown. B220 expression should be shown on both pro-B cell lines. Also, from the microarray analysis, a huge number of genes is differentially expressed between the Pro-B(+) and the por-B(-) cell lines (4000 are different by more than 2 fold. This could perhaps be discussed.

2) The quantification of absolute cell numbers should be shown in all culture experiments in addition of the frequency of T and B-lineage cells. This is would help understand whether Lmo2 has effects on differentiation, proliferation, or cell death. The authors state that both cell lines grow robustly on OP9, and that the pro-B(-) cells dies in the presence of Notch ligands. However, this is not shown. Quantification of absolute cell numbers, proliferation (e.g. CFSE dilution) and cell death (e.g. Annexin staining) of the cell line should be shown in both conditions in Figure 1. Of note, Figure S3 suggests that both cell lines die on OP9Dl4. Is that correct?

3) As the authors state, it remains unclear whether Lmo2 plays an important role before T-lineage commitment. This would greatly increase the impact of the manuscript if the authors could investigate this key point. For example, they could use CRISPR mediated deletion of Lmo2 in hematopoietic precursors (as they do in pro-B cells) and investigate their ability to generate T cells on OP9Dl4.

*Reviewer #3:*

1) This is a review of 'Lmo2 is essential to maintain the ability of progenitors to differentiate into T-cell lineage'.

The study by Hirano et al. addresses the mechanisms that reinforce the T-lineage potential at the B versus T-lineage branching, which is clearly dependent on NOTCH1 signaling. The authors took a genetic approach using Ebf1-deficient pro-B cells and exploited niche-based assays in which cells were either co-cultured on OP9 stromal cells or OP9-DL4 expressing the NOTCH1 ligand to isolate two Pro-B cell lines, Pro-B(+) cells and Pro-B(-) cells with and without T-lineage potential, respectively. An unbiased approach using global analysis of differentially expressed genes led to three candidates that, upon ectopic expression in Pro-B(-) cells, were either toxic or did not confer T-potential. The authors next took a hypothesis-driven approach based on LMO2 which is a major oncogene in T-cell acute lymphoblastic leukemia (T-ALL) and has been shown to reprogram differentiated blood cells or fibroblasts into hemopoietic stem cells. Upon finding that Lmo2 was more highly expressed in Pro-B(+), they either overexpressed Lmo2 in Pro-B(-) cells or deleted Lmo2 in Pro-B(+) cells via CRISPR-Cas9 to demonstrate a role for Lmo2 in conferring or maintaining the T-lineage potential. Mechanistically, Lmo2 is required for cell survival via the Bcl11a/Bcl2 axis while driving the T-lineage potential via Tcf7. Interestingly, Tcf7 together with Bcl2 are sufficient to confer T-lineage potential to pro-B(-) cells. Finally, LMO2 binds the upstream enhancer of the Tcf7 locus and controls the methylation status of the CpG island near the transcription start site of Tcf7 to secure a transcriptionally poised chromatin state required for NOTCH1 responsiveness.

2) The extensive genetic approach together with well-defined niche-based assays for quantitative assessments of gene function represent the major strength of the present study. The kinetics of cell survival and cell surface expression on OP9 stromal cells versus OP9-DL4 allowed for a clear measure of cellular response to NOTCH1 signaling with regards to T-lineage differentiation. While there is a clear phenotypic difference between Pro-B(+) and Pro-B(-) cells on DL4, data illustrated in Figure 1A indicate that Pro-B(+) cells are mostly at the DN2 stage is response to DL4 exposure. Similarly, overexpression of LMO2 in Pro-B(-) cells allows these cells to progress to the DN2 stage, not DN3 (Figure 1D). The evidence that Pro-B(+) cells progress to the DN3 stage as stated by the authors for data shown in Figure 1A remains to be documented. In addition to surface markers, molecular evidence for Trb gene rearrangement, Ptcra or Trgv3 expression would more convincingly support the proposition that these cells have indeed progressed to the DN3 stage.

3) The combination of unbiased identification of differentially expressed genes and of a hypothesis-driven approach is interesting and led to the identification of Lmo2 as driver gene and of Tcf7 as the target effector gene that determines NOTCH1 response. The microarray data served the purpose of identifying genes that are differentially expressed between Pro-B(+) and Pro-B(-) cells. Nonetheless, the selection of three genes for functional validation within this gene set was guided by previous knowledge on gene function. A more global and complementary approach by gene set enrichment analysis or principal component analysis with control cell types would have provided further insight on cell stage (ProT, ETP, DN2, ProB, etc) and would have brought to the authors' attention that column headings pertaining to Pro-B(+) and (-) were inadvertently swapped in Supplementary File 1. Nonetheless, the confirmatory RT-PCR shown in Figure 1B and 3A are consistent with the text and the conclusions.

4) Another question that arises from closer look at the microarray data as reported in Supplementary File 1 is the 643 fold difference in Xist gene expression between Pro-B(-) and pro-B(+) cells. 'This gene is expressed exclusively from the XIC of the inactive X chromosome' (IMMGEN) raising the concern that some of the differences within the differentially expressed gene set could inadvertently be due to male/female differences. The origin of these two cell lines with regards to the age and sex of donor mice should be clarified and discussed in the context of the observed data.

5) It is interesting that enrichment analysis with HOMER of DNA motifs that are associated with LMO2 peaks are RUNX, ETS and bHLH consensus binding sites. While HOMER may associate common transcription factors to these motifs as illustrated in Figure 6A, a previous genome-wide study of a transcription factor heptad in hematopoietic progenitor cells would be more relevant to the current study (PMID: 20887958 – DOI: 10.1016/j.stem.2010.07.016.) That LMO2 peaks in the present study overlaps with the same three DNA motifs found in a previous study in which LM02 is described as part of a transcription factor heptad that includes RUNX1, TAL1/SCL and LYL1 as bHLH factors, and two ETS family members underlines the quality of the ChIP-seq data reported here. Second, it would be interesting to address the question whether the same motifs were enriched in the differentially expressed gene set between Pro-B(+) and Pro-B(-) cells, to further consolidate the evidence that the endogenous Lmo2 controls the difference between the two cell lines.

6) LMO2 is a major oncogene in T-ALL. Nonetheless, LMO2 expression was found to be elevated in a high proportion of diffuse large B cell lymphomas (DLBCL) in which LMO2 regulates genes implicated in kinetochore function, chromosome assembly, and mitosis (PMID: 22517897 – DOI: 10.1182/blood-2012-01-403154). Moreover, Lmo2 expression is also elevated in B cells from normal spleen and lymph nodes as can be quieried in the IMMGEN database. The present study is more likely to reveal Lmo2 and Tcf7 function in the absence of Ebf1 in Pro-B cells. The rescue of Pro-B(-) by Lmo2 for their capacities to survive and progress to the DN2 stage on OP9-DL4 is convincing. To address the consequence of restoring Ebf1 expression in this context would add further insight into the mechanism regulating NOTCH1 response, as well as the combinatorial interaction between cell autonomous transcription factors and signaling pathways that define cell fate.

1) Nuclear staining for TCF1 and GATA3 shown in Figure 4B is convincing. Unfortunately, LMO2 nuclear staining to demonstrate the difference between the two cell lines are not as convincing;

'We found that the expression levels of Lmo2 mRNA and protein were ~3 fold higher in pro-B(+) than pro-B(-) cells (Figure 1B and C)'. RT-PCR shown in the source data of Figure 1B are consistent with a 3-fold difference in mRNA expression levels as normalized to Actb levels. However, there is no comparative analysis of LMO2 protein levels by flow cytometry and it is not clear how the modest shift in fluorescence was estimated as three-fold higher in Pro-B(+) cells.

2) Please clarify the reference for NICD and RUNX1 ChIP-seq in the result section (Supplementary Figure 6). The reference was given in the discussion, which is confusing because the dataset was used to highlight LMO2 and RUNX1 binding, while only LMO2 ChIP-seq was performed in the present study.

3) Data shown in Figure 5 should be better explained and clarified in the Figure legend. What does the matrix represent?

4) Figure 2B: sgLmo2 introduction into Pro-B(+) cells abolished LMO2 protein expression on day 4 after sgRNA transduction. Yet on day 5, when cells were transferred on OP9-DL4 for another 3 days, these cells were still able to progress to the DN2 stage compared to cells that were maintained on OP9 and remained DN1. How do the authors explain the fact that abolishing LMO2 protein levels on day 4 had no consequence on day 5, when cells were transferred to OP-DL4 for 3 days? That deficient progression to the DN2 stage should be detectable only after 10 days on OP9 plus another 3 days on OP9-DL4 is a very slow response (Figure 2B). The authors conclude on a 'slower time-scale transcriptional changes, including histone modifications, chromatin remodeling, and DNA methylation.' Have the authors verified the kinetics of either histone marks of the Tcf7 locus or of TCF1 protein expression by flow cytometry during this 5- to 10-day maintenance on OP9, and after a 3-day exposure to OP9-DL1?

5) Please note the convention for murine and human gene and protein:

Lmo2 (italics) : murine gene

LMO2 (italics): human gene

LMO2 : murine or human protein

---

## [Author Response]

Essential revisions:1) A better characterization of how the proB cells were derived.2) A characterization of the differentiation potential of the proB cells cultured on OP9-DL4 that shows their ability to go beyond the DN2 stage of T cell development.

Thank you so much for raising these points. We have added new Figure 1—figure supplement 1 A, B C, D, E and F to clarify the points. We showed that TD7 is able to support B cell differentiation (A), generation of higher % of lineage marker-positive cells from FL progenitors on TD7 (B), expression levels of B220 in pro-B(-) and pro-B(+) cells (C), differentiation potential of pro-B(+) into DP stage in vitro (D) and mature CD4 and CD8 T cells in vivo (E), and expression levels of Notch receptors in pro-B(-) and pro-B(+) cells (F). We have added explanation for these results in the Results section.

3) Demonstration of the timing of histone modifications after CRISPR/Cas9 induced loss of Lmo2 expression, which would help explain the observed delay in the loss of T cell potential.

Thank you for raising this important point. First, we have checked expression levels of *Tcf7* transcripts after *Lmo2* disruption, and found that down-regulation of *Tcf7* expression was only observed at 10 days after sgLmo2-transduction (new Figure 4—figure supplement 1). We also carried out ChIP-qPCR analysis for H3K4-3Me, one of the active histone marks, around the *Tcf7* locus. Our new results suggest that the transcriptional start site of the *Tcf7* has highly enriched H3K4-3Me and levels of this modification were gradually decreased (modestly on day 5, and significantly on day10) after loss of *Lmo2* expression. This data was added in Figure 5-fugure supplement 1 and described in the Results section.

4) Cell death calculations need to be redone as % cell death.

Thanks for bringing this up. We have added a graph for % cell death in the new Figure 3—figure supplement 1 as you mentioned, and changed description about this point in the Results section as, ‘Transduction of *Bcl2* significantly protected pro-B(-) cells from cell death, before and after Notch stimulation’.

Reviewer #1:The work by Hirano et al. addresses an important and not fully resolved issue regarding the regulation of the initiation of T cell lineage commitment. The approach used employs an Ebf1-deficient pro-B cell line system that can be induced to differentiate toward the T cell lineage after receiving strong Notch signaling in vitro. However, a subset of these pro-B cells failed to give rise to T-lineage cells, which was attributed to low levels of Lmo2 expression as compared to the pro-B cells that retain T-lineage differentiation potential. Enforced expression of Lmo2 rescued the defect, which was shown to act by regulating epigenetic changes at the Tcf7 gene locus. However, what remains unclear is whether Lmo2 is required at the start of T-cell lineage differentiation to enable entry into the program, as well as serving to facilitate the elaboration of the program as the cells commit to the T cell lineage. Nevertheless, even without clarity as to the cellular subset or subsets requiring Lmo2 function, the present finds provide a clear rationale for considering Lmo2 as an important player during early T cell differentiation.A few concerns, when addressed, would improve the impact of the findings.1. It is clear that the proB(-) do not fare well in the Dll4+ cell cultures, and that the proB(+) cells can differentiate and survive well in the Dll4+ cell cultures, however, with only CD44 and CD25 expression as the only indicators of differentiation, and CD44 is expressed on both culture conditions, it would be good to also show an additional T-lineage related marker, e.g., CD90 to further support their differentiation as T cells. Additionally, do the proB(+) cells differentiate to later stages of T cell development, ie., do they eventually express CD3, CD4, CD8 and TCRs?

Thank you for this point. In new Figure 1-supplement 1 D and E, we have added results to show the differentiation potential of pro-B(+) cells into DP stage in vitro (D), and mature CD4 and CD8 T cells in vivo (E).

2. The results shown in Figure 2 make it seem that in the acute absence of Lmo2 the effect is not in the initiation of T-lineage commitment and early differentiation but rather affecting their progression or transition to later stages. Perhaps the authors should consider this interpretation of their results in the discussion and/or Results section.

In the previous results, we had used Cas9-GFP^high^ pro-B(+) cells to delete *Lmo2*, however, the majority of them died after Notch stimulation, because of toxicity of high Cas9 expression. Thus, we had analyzed minor survived cells for the previous results, and it messed the flow profiles up. To solve this problem, we established pro-B(+) cell lines with intermediate Cas9-GFP expression. We confirmed that *Lmo2*-deficient pro-B(+) cells with intermediate Cas9 expression still had the significantly abrogated T-lineage potential with clearer flow profiles, which strongly supports our conclusion. These results were replaced in new Figure 2B.

3. The Supplemental Excel file with the microarray results appears to be missed labeled, ie., reversed heading for ProB(-) vs ProB(+), as the data shown are the opposite to what is summarized in Figure 1.

Thank you for pointing this out. We have fixed the labels in the new Supplemental Excel file.

4. The survival/death analysis shown in Figure 3, supplement 1, appears to show an increase in cell death in proB(-) cells when cultured on Dll4+ cells, while the ectopic expression of Bcl2 would appear to lessen the number of dying cells. However, if one were to use the ratio of dead cells from Dll4 vs Mock cell cultures (Dll4/Mock) to determine the fold increase in the % of cells dying in each condition, then the number would be as follows: 2x for proB(+); 6.4x for proB(-)/Mock; and 7.3x for proB(-)/Bcl2, which would lead to the conclusion that Bcl2 is not rescuing the cells from dying, and I feel that this is the more likely and correct interpretation of the results. Minimally, the authors should convey this notion in the results.

Thanks for bringing this up. We have added a graph for % cell death in the new Figure 3—figure supplement 1, and changed description about this point in the Results section as, ‘Transduction of *Bcl2* significantly protected pro-B(-) cells from cell death, before and after Notch stimulation’.

5. In Figure 4c, the results are quite striking and beautifully demonstrate that ectopic expression TCF1 in proB(-)/Bcl2 cells enables their ability to differentiate into CD25+ cells. However, it is not clear whether this would require Dll4-dependent signals as the Mock cell cultures are not shown. The authors should include this experiment, as it provides insights as to the requirement for Notch signaling in cells that have enforced TCF1 expression, which have been shown by the Bhandoola lab and others.

Thank you for the important suggestion. In our experimental system, we have not been able to see development of T cells (generation of CD25+ cells) in the TCF1 introduced pro-B(-) cells without Dll4 stimulation. We introduced mouse *Tcf7* using lentivirus, and it would have lower TCF1 expression levels than the retrovirus vector for human *TCF7*, the Bhandoola’s group used. We have included the flow data without Dll4 stimulation (OP9-Mock) in the new Figure 4C.

6. Lastly, some of the discussion statements, in page 18, which argue for a role for Lmo2 in prethymic progenitors are not fully backed by the results presented in the paper, as prethymic progenitors were not examined, these conclusions should be tempered as possible rather as having been shown.

We agree with the reviewer and fixed this point in the Summary, Introduction, Result and Discussion sections. We have changed the description in the text from “pre-thymic progenitors” to “T-cell progenitors”.

Reviewer #2:In this manuscript, Hirano et al. investigate the mechanisms that maintain T cell potential in pre-thymic lymphoid precursors. The authors establish a new stromal cell lines from fetal thymus, called TD7. They use this cell line to establish an Ebf1-deficient pro-B cell line (called pro-B(-)), using a protocol similar to the one previously described to generate Ebf1-deficient pro-B cells using the well-established OP9 stromal cell line (called pro-B(+)). Surprisingly, they find that unlike pro-B(+), pro-B(-) fail to engage toward the T cells lineage in response to Notch signaling. To understand this puzzling difference, they transcriptionally profile the two pro-B cell lines, and find that the transcription factor Lmo2 is expressed at lower level on pro-B(-) compared to pro-B(+). They propose that Lmo2 plays important functions to enable expression of the transcription factor TCF-1 in response to Notch signaling (encoded by Tcf7). To support this model, they notably show that Lmo2 overexpression rescues the T cell potential of pro-B(-), and conversely, Lmo2 loss of function restrain the T cell potential of pro-B(+). Importantly, they show that TCF-1 ectopic expression rescue the T cell potential of pro-B(-). They propose that Lmo2 may play key functions in maintaining T cell potential in pre-thymic T cell precursors.In this elegant study, the authors provide a detailed characterization of the effect of ectopically expressing or deleting Lmo2 on transcription and epigenetics in Ebf1-deficient B cell lines, and their differentiation toward the T cell lineage. The results provided likely have an important impact for our understanding of the mechanisms driving the development of Lmo2-induced T cell leukemia. However, it is unclear whether this model also apply to normal hematopoiesis.I believe that this very interesting study would be greatly strengthened and its impact increased if the authors could address the following points:1) The novel TD7 stromal cell line and the pro-B cells generated on this cell line are poorly characterized. The authors state that "We established our original thymic stromal cell line, TD7, in which HSCs differentiate into CD19^+^ B-lineage cells in the presence of IL-7 or into Thy1+CD25+ T lineage cells upon receiving Notch signaling.", "the establishment of pro-B cells on TD7 was accompanied by a prolonged emergence of myeloid marker-positive cells compared to that on OP9, and the remaining pro-B cells lost their pluripotency". However, this data is shown. The careful characterization of the cells generated on TD7 would be very valuable for the readers, and key to properly interpret the data presented in the manuscript. It is presently unclear how the pro-B(-) are related to pro-B cells. The authors mention that pro-B(-) cells express B220 but this is not shown. B220 expression should be shown on both pro-B cell lines. Also, from the microarray analysis, a huge number of genes is differentially expressed between the Pro-B(+) and the por-B(-) cell lines (4000 are different by more than 2 fold. This could perhaps be discussed.

Thank you for the encouragement to clarify this point. We showed that TD7 is able to support B cell differentiation, higher % of lineage marker-positive cells from FL progenitors is generated on TD7, and expression levels of B220 in pro-B(-) and (+) cells in the new Figure 1—figure supplement 1A, B and C. We have also analyzed differentially expressed genes in pro-B(+) and pro-B(-) cells, and found that these genes were enriched for genes related to ‘Hematopoietic cell lineage’ in KEGG pathway analysis. We have added this new results in the Figure 1—figure supplement 1G and described in the Results section.

2) The quantification of absolute cell numbers should be shown in all culture experiments in addition of the frequency of T and B-lineage cells. This is would help understand whether Lmo2 has effects on differentiation, proliferation, or cell death. The authors state that both cell lines grow robustly on OP9, and that the pro-B(-) cells dies in the presence of Notch ligands. However, this is not shown. Quantification of absolute cell numbers, proliferation (e.g. CFSE dilution) and cell death (e.g. Annexin staining) of the cell line should be shown in both conditions in Figure 1. Of note, Figure S3 suggests that both cell lines die on OP9Dl4. Is that correct?

Thank you for raising this important point. Absolute cell numbers are totally dependent on the input cell numbers, thus we indicated numbers of CD25+ cells as fold expansion (/input cell number) in Figure 1A and D. We also show absolute numbers of CD25+ cells in Lmo2 -deficient pro-B(+)cells in Figure 2B.

Both pro-B(-) and pro-B(+) cells have slower proliferation with more dead cells after Notch stimulation. Thus, you are right, both cell lines die on OP9-Dll4.

3) As the authors state, it remains unclear whether Lmo2 plays an important role before T-lineage commitment. This would greatly increase the impact of the manuscript if the authors could investigate this key point. For example, they could use CRISPR mediated deletion of Lmo2 in hematopoietic precursors (as they do in pro-B cells) and investigate their ability to generate T cells on OP9Dl4.

Thank you so much for the suggestion. We have performed experiments as the reviewer suggested. Hematopoietic progenitor cells from BM of Cas9 knock-in mouse were transduced with sgLmo2, and they were cultured without OP9 for two days (if we culture them with OP9-Mock, they will differentiate into B-lineage). Then, they were transferred onto a OP9-Dll1 monolayer and cultured for 4 days. However, the sgLmo2-introduced progenitors were able to differentiate into DN2 stage, as new Figure 2—figure supplement 1A and B. Our results using pro-B(+) with acute deletion of *Lmo2* indicate that the loss of Lmo2 induces formation of closed chromatin structure at the *Tcf7* locus, but it would take ~10 days. In our knowledge, it is really difficult to maintain sgLmo2-introduced hematopoietic precursors in vitro for 10 days before Notch stimulation. We added these results in the new Figure 2—figure supplement 1A and B, and described in the Results section.

Reviewer #3:[…] 1) Nuclear staining for TCF1 and GATA3 shown in Figure 4B is convincing. Unfortunately, LMO2 nuclear staining to demonstrate the difference between the two cell lines are not as convincing;'We found that the expression levels of Lmo2 mRNA and protein were ~3 fold higher in pro-B(+) than pro-B(-) cells (Figure 1B and C)'. RT-PCR shown in the source data of Figure 1B are consistent with a 3-fold difference in mRNA expression levels as normalized to Actb levels. However, there is no comparative analysis of LMO2 protein levels by flow cytometry and it is not clear how the modest shift in fluorescence was estimated as three-fold higher in Pro-B(+) cells.

Thanks for bringing this up. We have added a graph to show the mean fluorescent intensity of Lmo2 protein in pro-B(-) and pro-B(+) cells in the new Figure 1C.

2) Please clarify the reference for NICD and RUNX1 ChIP-seq in the result section (Supplementary Figure 6). The reference was given in the discussion, which is confusing because the dataset was used to highlight LMO2 and RUNX1 binding, while only LMO2 ChIP-seq was performed in the present study.

Thank you for pointing this out. We have added information of RBPJ and Runx1 ChIP-seq data (GEO numbers and references) in the Result section.

3) Data shown in Figure 5 should be better explained and clarified in the Figure legend. What does the matrix represent?

Thank you for the encouragement to clarify this. We have replaced the figure legend with clearer explanation.

4) Figure 2B: sgLmo2 introduction into Pro-B(+) cells abolished LMO2 protein expression on day 4 after sgRNA transduction. Yet on day 5, when cells were transferred on OP9-DL4 for another 3 days, these cells were still able to progress to the DN2 stage compared to cells that were maintained on OP9 and remained DN1. How do the authors explain the fact that abolishing LMO2 protein levels on day 4 had no consequence on day 5, when cells were transferred to OP-DL4 for 3 days? That deficient progression to the DN2 stage should be detectable only after 10 days on OP9 plus another 3 days on OP9-DL4 is a very slow response (Figure 2B). The authors conclude on a 'slower time-scale transcriptional changes, including histone modifications, chromatin remodeling, and DNA methylation.' Have the authors verified the kinetics of either histone marks of the Tcf7 locus or of TCF1 protein expression by flow cytometry during this 5- to 10-day maintenance on OP9, and after a 3-day exposure to OP9-DL1?

Thank you for raising this important point. First, we have checked expression levels of *Tcf7* transcripts after *Lmo2* disruption, and found that down-regulation of *Tcf7* expression was only observed at 10 days after sgLmo2-transduction (new Figure 4—figure supplement 1). We also carried out ChIP-qPCR analysis for H3K4-3Me, one of the active histone marks, around the *Tcf7* locus. Our new results suggest that the transcriptional start site of the *Tcf7* has highly enriched H3K4-3Me and levels of this modification were gradually decreased (modestly on day 5, and significantly on day10) after loss of *Lmo2* expression. This data was added in Figure 5-fugure supplement 1 and described in the Results section.

5) Please note the convention for murine and human gene and protein:Lmo2 (italics) : murine geneLMO2 (italics): human geneLMO2 : murine or human protein

Thank you for this correction. We have carefully changed them.